# Bacterial N4-methylcytosine as an epigenetic mark in eukaryotic DNA

Fernando Rodriguez [1,3], Irina A. Yushenova [1,3], Daniel DiCorpo [1,2] & Irina R. Arkhipova [1✉]

DNA modifications are used to regulate gene expression and defend against invading genetic elements. In eukaryotes, modifications predominantly involve C5-methylcytosine (5mC) and occasionally N6-methyladenine (6mA), while bacteria frequently use N4-methylcytosine (4mC) in addition to 5mC and 6mA. Here we report that 4mC can serve as an epigenetic mark in eukaryotes. Bdelloid rotifers, tiny freshwater invertebrates with transposon-poor genomes rich in foreign genes, lack canonical eukaryotic C5-methyltransferases for 5mC addition, but encode an amino-methyltransferase, N4CMT, captured from bacteria >60 Mya. N4CMT deposits 4mC at active transposons and certain tandem repeats, and fusion to a chromodomain shapes its "histone-read-DNA-write" architecture recognizing silent chromatin marks. Furthermore, amplification of SETDB1 H3K9me3 histone methyltransferases yields variants preferentially binding 4mC-DNA, suggesting "DNA-read-histone-write" partnership to maintain chromatin-based silencing. Our results show how non-native DNA methyl groups can reshape epigenetic systems to silence transposons and demonstrate the potential of horizontal gene transfer to drive regulatory innovation in eukaryotes.

---

[1] Josephine Bay Paul Center for Comparative Molecular Biology and Evolution, Marine Biological Laboratory, Woods Hole, MA 02543, USA. [2] Present address: Department of Biostatistics, Boston University School of Public Health, Boston, MA 02118, USA. [3] These authors contributed equally: Fernando Rodriguez, Irina A. Yushenova. ✉email: iarkhipova@mbl.edu

Modification of nucleobases without changes in the underlying genetic code offers unmatched opportunities for "writing" extra information onto DNA, the primary carrier of hereditary material. Covalent association of modifying groups with DNA provides advantages over more easily removable carriers of epigenetic information, such as RNA or proteins, for potential transmission across cell divisions and generations. In bacteria and archaea, DNA modifications are first and foremost associated with restriction–modification (R–M) systems acting to discriminate and destroy the invading foreign DNA, although multiple "orphan" methyltransferases (MTases) may perform regulatory functions[1,2]. Eukaryotes mostly use base modifications for regulatory purposes, with the predominant form of epigenetic modification in eukaryotic genomes being C5-methylcytosine (5mC) and its derivatives[3,4]. Often called "the fifth base", 5mC plays an important role in genome defense against mobile genetic elements, and is often associated with transcriptional silencing, establishment of the closed chromatin configuration, and repressive histone modifications[5]. The 5mC mark is introduced by C5-MTases, DNMT1 and DNMT3, thought to have originated from bacterial C5-MTases in early eukaryotes via fusions with additional domains interacting with proteins and DNA[6], while DNMT2 acts primarily on tRNA[7,8]. Recently, another modified base, N6-methyladenine (6mA), gained attention as a possible novel form of epigenetic modification in diverse eukaryotes, although its role remains controversial[9–11]. In 6mA, a methyl group is added to an exocyclic amino group of adenines by amino-MTases, some of which are related to RNA-modifying MTases[12,13]. However, the third type of DNA methylation naturally occurring in bacteria, N4-methylcytosine (4mC), has not been demonstrated to act as an epigenetic mark in eukaryotes, and most claims of eukaryotic 4mC lack confirmation by orthogonal methods and do not identify the enzymatic component[14]. Here, we combine multiple lines of evidence to establish that 4mC modification can be recruited as an epigenetic mark in eukaryotic genomes, and to characterize the underlying enzymatic machinery. We focus our attention on epigenetic silencing phenomena that involve DNA and histone modifications, without expanding into broader areas involving nuclear organization or post-transcriptional silencing. Our work demonstrates how a horizontally transferred gene can become part of a complex regulatory system maintained by selection over tens of millions of years of evolution.

## Results

**A bacterial amino-MTase in bdelloid rotifers.** Rotifers of the class Bdelloidea are tiny freshwater invertebrates a fraction of a millimeter long, characterized by clonal reproduction, eutely, direct development, syncytial tissues, and paleotetraploid genome structure[15]. They are known for an unmatched ability to incorporate foreign genes into genomic DNA, largely preserving their functionality[16]. In sequenced bdelloids, 8–12% of coding sequences are of non-metazoan, mostly bacterial, origin[17–19]. Surprisingly, we found that one such bacterial gene in the sequenced bdelloid *Adineta vaga*[17] is represented by an allelic pair of MTases containing the N6_N4_MTase domain (PF01555), which is closely related to amino-MTases of bacterial R–M systems acting on the exocyclic amino group of adenines and cytosines (Fig. 1a; Supplementary Fig. 1). Its orthologs, sharing the same five conserved intron positions, are present in sequenced representatives of each major family of the class Bdelloidea, dating back 40–60 Mya, but are absent from sequenced members of the sister class Monogononta or from other sequenced eukaryotes (Fig. 1e, f; Supplementary Fig. 2). Both classes, however, encode amino-MTases implicated by various authors in adding 6mA marks to eukaryotic DNA: METTL4-like (PF05063:

MT-A70), N6AMT1-like (PF05175: *MTS*) and N6AMT2-like (PF10237: *N6-adenineMIase*)[12,20–23] (Fig. 1b, f). Notably, none of the sequenced rotifers harbor *Dnmt1* or *Dnmt3* genes for the most common eukaryotic C5-MTases, encoding only the tRNA-modifying *Dnmt2/Trdmt*.

The N6_N4_MTase found in *A. vaga* belongs to the permuted type, in which the catalytic domain is located N-terminally to the S-adenosylmethionine (AdoMet) binding domain[24] (Fig. 1a). Its evolutionary history and taxonomic distribution (Fig. 1e, f) differs dramatically from that of 5mC- or N6A-MTases[6]. The small non-permuted pan-eukaryotic MTases N6AMT1 and N6AMT2 (Fig. 1b), variably annotated either as N(6)-adenine MTases or as small N5-glutamine (HemK-like) and lysine (eEF1A) MTases, respectively, have been implicated in N6A methylation based on knockout/knockdown data[21,25], but do not carry N- or C-terminal extensions, and modify proteins rather than DNA in functional assays[26–30], suggesting that in vivo perturbations may have indirect effects. The presumptive N6A-MTase METTL4, which in *Drosophila* adds m6A to U2 snRNA in vitro[31], has a conserved N-terminal domain (KOG2356: transcriptional activator, adenine-specific DNA methyltransferase) present in METTL4-like ORFs of most eukaryotes, including *A. vaga* (Fig. 1b, top). This permuted MTase, found in all bdelloids, may have persisted in eukaryotes throughout their evolutionary history (Fig. 1f, Supplementary Table 1). In contrast, the bdelloid N6_N4_MTase has no eukaryotic homologs, and can be aligned only with permuted bacterial N4C- and N6A-MTases (Type II, subtype β), which cluster in accordance with their target recognition domains (TRD) recognizing specific targets compiled in REBASE[1,24,32] (Fig. 1e; Supplementary Data 1). Interestingly, the bdelloid lineage clusters with phage MTases of unknown target specificity, and its closest bacterial homologs are N4C-MTases recognizing TCGA and CCSGG. Thus, we tentatively assigned it to N4C-MTases and named it N4CMT, since it harbors the catalytic SPPY motif shared with most bacterial N4C-MTases and differing from bacterial N6A-MTases (DPPY), eukaryotic N6AMT1 (NPPY), N6AMT2 (DPPY/F) or METTL4-like enzymes (DPPW, also seen in METTL3/IME4-like m6A-RNA MTases)[12,24,33] (Supplementary Table 2).

**Presence of 4mC and 6mA marks in genomic DNA.** We next sought to find out whether recruitment of a horizontally transferred bacterial MTase resulted in the establishment of bacterial epigenetic marks in bdelloid genomic DNA (gDNA). A strong indication that N4CMT could interact with chromatin to add 4mC to gDNA comes from the presence of a eukaryotic chromodomain from the HP1/chromobox subfamily of methylated lysine-binding Royal family of structural folds[34] at the C-terminus of the bacterial *N6_N4_MTase* moiety in sequenced bdelloids (Fig. 1a, Supplementary Fig. 2).

To detect 4mC/6mA marks in bdelloid genomes, we extracted gDNA from the *A. vaga* laboratory reference strain (hereafter Av-ref)[17] fed with methyl-free *Escherichia coli* (Supplementary Table 3), and performed immuno-dot-blotting with anti-4mC and anti-6mA antibodies (Methods). We also extracted gDNA from the natural *A. vaga* isolate L1 (hereafter AvL1; Supplementary Movie; Fig. 1g), which was caught in the wild and identified as *A. vaga* through morphological criteria and mtDNA phylogeny, but represents a distinct morphospecies within the *A. vaga* species complex, as its gDNA is only 88% identical to Av-ref[35]. Figure 1c shows that gDNA from Av-ref and AvL1 reacts positively with both antibodies, suggesting the presence of 4mC and 6mA marks. Control DNAs isolated from the *dam-/dcm-*, DH5α and Top10 *E. coli* strains, or from *E. coli* M28 strain used as food (Supplementary Table 3), did not react with anti-4mC antibodies (Fig. 1c), and neither we observe cross-reactivity of the

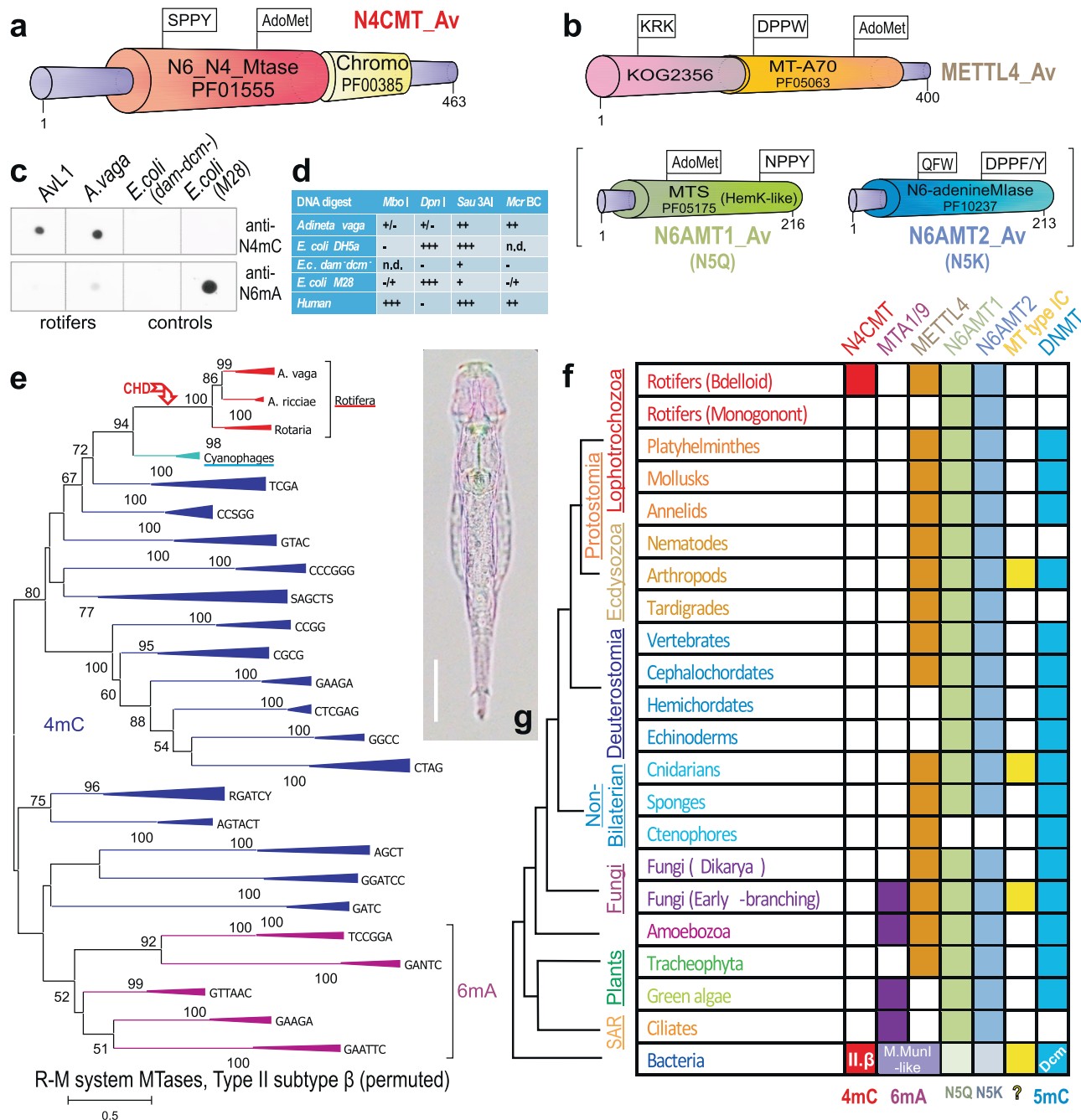

**Fig. 1 Putative DNA methyltransferases and modified bases in bdelloid rotifers. a, b** Domain structure of putative N4C (**a**) and N6A (**b**) bdelloid amino-MTases. PFAM/KOG domains are indicated; conserved catalytic motifs and S-adenosylmethionine (AdoMet) binding sites are flagged; numbers correspond to aa positions in *A. vaga*. See Supplementary Data 1 for gene IDs and aa sequences. **c**, Immuno-dot-blot analysis of membrane-immobilized gDNA from *A. vaga* Av-ref (746 ng), AvL1 (500 ng), *E. coli* C2925 *dam−/dcm−* (550 ng) and *E. coli* M28 (2 µg) probed with anti-4mC (top panel) and anti-6mA (bottom panel) antibodies. **d**, Summary of gDNA digestion (+) with restriction enzymes differing by methylation sensitivity: *Mbo*I (blocked by *dam* methylation); *Dpn*I (cleaves only *dam* methylated DNA); *Sau*3AI (not sensitive to *dam* or *dcm* methylation); *Mcr*BC (cleaves at any methylated cytosine). **e** Neighbor-joining phylogram of permuted MTases of Type II, subtype β, displaying clustering by recognition sequences obtained from REBASE. Clustering is not intended to uncover phylogenetic relationships in bacteria. Red arrow indicates acquisition of a chromodomain (CHD) by the bdelloid N4CMT. Sequences are provided in Supplementary Data 1. **f** Phyletic distribution patterns of putative DNA methyltransferases implicated in 4mC, 6mA, and 5mC addition. A consensus cladogram of metazoan phyla is shown on the left. **g** *Adineta vaga* AvL1 under polychromatic polarizing microscope. Photo credit: M. Shribak, I. Yushenova. Scale bar, 50 µm.

anti-4mC antibody with 5mC-containing human DNA. Also consistent with the presence of modified cytosines were the results of treatment of total *A. vaga* gDNA with the McrBC endonuclease, which cleaves at any methylated cytosine (5mC, 5hmC, 4mC)[36,37] (Fig. 1d; see also Fig. 3b below). Together with

the absence of C5-MTases, the similarity of N4CMT to bacterial N4C-MTases (Fig. 1e), and the lack of 5mC deamination signatures in gDNA from observed/expected CpG ratios (Supplementary Fig. 3a), our data support the hypothesis that cytosines in bdelloids are modified at the N4- rather than C5-

positions. Still, signals in gDNA may originate from residual methylated bacterial DNA from sources other than food. Thus, we sought to examine the distribution of 4mC marks over annotated genomic features in bona fide eukaryotic contigs.

**Genome-wide analysis of 4mC and 6mA by DIP-seq.** We exploited immunoreactivity of bdelloid DNA with anti-4mC and anti-6mA antibodies to assess the genome-wide distribution of these methylation marks by DIP-seq (DNA immunoprecipitation followed by sequencing, also called MeDIP-seq; see Methods). After read mapping to Av-ref, MACS peak-calling tool identified 1008 and 1735 DIP-seq peaks (p-value < 1e−5) for 4mC and 6mA, respectively, which were broadly distributed throughout the assembly. To uncover biologically relevant patterns behind peak distribution, we compared average coverage densities for 4mC and 6mA near annotated genomic features, such as gene coding sequences (CDS) and transposable elements (TEs). We visualized the distribution of 4mC and 6mA sites near TEs by aligning TEs at the 5′ end (profiles) or aligning TE bodies from 5′ to 3′ end at a fixed distance (metaprofiles), and plotting the IP occupancy, which shows the relative number of DIP-seq reads against the total number of TEs for each bin size within a pre-determined upstream and downstream window. DIP-seq data for 4mC show elevated density near TE insertions in comparison with 6mA (Fig. 2a, left and right), suggesting that TE insertions could be an important 4mC modification target. For gene annotations, IP occupancy (Fig. 2a, center) does not show an increase in density at the transcription start site (TSS) seen in TEs. After peak calling with MACS, we also compared relative numbers of peaks with intersected annotations: about one-half of 4mC peaks (468 out of 1008) and a quarter of 6mA peaks (430 out of 1735) are close to TEs, and more 6mA peaks (1261 out of 1735) than 4mC peaks (398 out of 1008) are close to gene annotations. Genometric spatial correlation analysis (Supplementary Note 1; Supplementary Table 4; Supplementary Fig. 4) further shows that DIP-seq 4mC marks are closer to TEs than would be expected from a uniform distribution.

The presence and distribution of 4mC and 6mA DNA modifications in the AvL1 strain were similarly interrogated by DIP-seq. We generated DIP-seq reads and mapped them onto AvL1 assembly (Methods). After peak calling with MACS, we identified 1473 and 1385 peaks (p-value < 1e−05) for 4mC and 6mA, respectively. To further understand methylation patterns in AvL1, we performed initial gene and TE annotations with fully automated training methods for gene prediction, using genomic and RNA-Seq data (Braker2; see Methods) (Supplementary Table 5). AvL1 repeat library was constructed de novo, manually curated, and used to annotate TEs (Methods). Initial analysis showed that 4mC-DIP-seq and 6mA-DIP-seq have similar distribution profiles in the assembly (Supplementary Fig. 5a, b) with enrichment of both marks towards genes and transposons, part of which may be due to the undetected presence of unknown types of low copy-number TEs in gene annotations. Nevertheless, cluster analysis showed an increase in 4mC being detected in a subset of transposons (clusters 1 and 2, Supplementary Fig. 5d). After peak calling, we found 1097 4mC peaks (out of 1473) and 1042 6mA peaks (out of 1385) close to TEs, while 863 4mC and 813 6mA peaks are close to genes (excluding TEs). Genometric correlation analysis in AvL1 showed that both 4mC and 6mA modification peaks (Supplementary Note 4; Supplementary Table 4; Supplementary Fig. 4) display a small absolute positive correlation with TEs, being closer than expected to TEs than to gene models as reference features (Jaccard and permutation test). Together, DIP-seq data in both Av-ref and Av-L1 suggest preferential localization of 4mC over TEs.

**Modification analysis at single-base resolution by SMRT-seq.** While immuno-dot-blots and differential gDNA digestion suggested the presence of 4mC in bdelloid gDNA, one cannot fully eliminate gDNA from commensal bacteria, even using methyl-free E. coli food strains and applying starvation/antibiotic treatments prior to DNA extraction (Methods). Hence, we chose not to use mass-spectrometry (MS) as a method to confirm the presence of 4mC in bdelloids, especially considering that unknown MS-peaks can comigrate with 4mC[14]. Further, the low resolution of the DIP-seq method limits the power of correlation analyses to the length of DNA fragments used for antibody binding (250–450 bp), not to mention residual IgG binding to non-modified fragments inherent to the method[38]. Thus, we chose to examine the genome-wide distribution of modified bases by single-molecule real-time (SMRT) sequencing, which provides single-nucleotide resolution and allows validation of metazoan/bacterial contigs (Methods).

SMRT-based detection exploits kinetic signatures of polymerase passage through modified vs non-modified bases and is quantified in terms of inter-pulse duration (IPD) ratios. It is best suited for the detection of 4mC and 6mA, characterized by strong kinetic signatures, which require ~10-fold lower coverage than 5mC detection (Pacific Biosciences Methylome Analysis Technical Note) and is widely used in bacterial methylome analyses[32,39]. We obtained PacBio reads (15 SMRT cells, totaling 9.87 Gb) from gDNA extracted from AvL1 eggs and analyzed the kinetic profiles with SMRT® Portal (Methods). Prior to quantification of modified bases, we bioinformatically removed residual bacterial contigs, which show high methylation density.

SMRT-analysis detected 4mC modifications on 21,016 cytosines (0.0643% of the total cytosines in the assembly) and 6mA modifications on 17,886 adenines (0.0236% of total adenines) using a minimum cutoff PacBio coverage defined in Fig. 2f (see Supplementary Table 6 for comparison of 10× and 20× coverage levels). As with DIP-seq, SMRT-seq shows a broad distribution of both modifications across AvL1 assembly. Comparison of DIP-seq and SMRT-seq modification patterns shows considerable overlap, with 36% of 4mC peaks and 32% of 6mA peaks overlapping with 4mC and 6mA identified by SMRT analysis, respectively, indicating that many peaks are conserved between eggs and adults. Following normalization of SMRT-seq methylation fraction values per modified base (see Methods), it is seen that 4mC and 6mA DIP-seq peak summits overlap with modified regions for PacBio 4mC and 6mA, respectively; plotting only 4mC-marked CpG sites shows a similar increase towards DIP-seq 4mC peaks (Supplementary Fig. 6a). The peak overlap is quite substantial, given the modest proportion of modified bases in the genome, and might reflect the general lack of methylation reprogramming during development in protostomes, known at least for 5mC[40].

In contrast to the predominantly symmetric patterns of 5mC deposition at CpG doublets in eukaryotes, AvL1 shows mostly asymmetric patterns of methylation for both 4mC and 6mA, i.e., only one strand is usually modified (Fig. 2b displays typical examples). At 4mC sites, CpG and CpA dinucleotides are the most prevalent, making up 74% of modified doublets. For better identification of sequence preferences, we extracted different sequence windows (5, 10, and 20 bp) upstream and downstream from 4mC sites and searched for significant motif enrichment with MEME-ChIP (Methods) (Fig. 2d). For 4mC, three motifs with CG or CA dinucleotides were most significantly enriched (from p = 2.8e−593 to p = 1.4e−513). For 6mA, a similar approach yielded three significantly enriched short motifs (from p = 7.3e−656 to p = 4.3e−420); increasing the motif length yielded GA embedded in an A-rich region (p = 2.4e−1243). However, none of these matched the RRACH motif found at

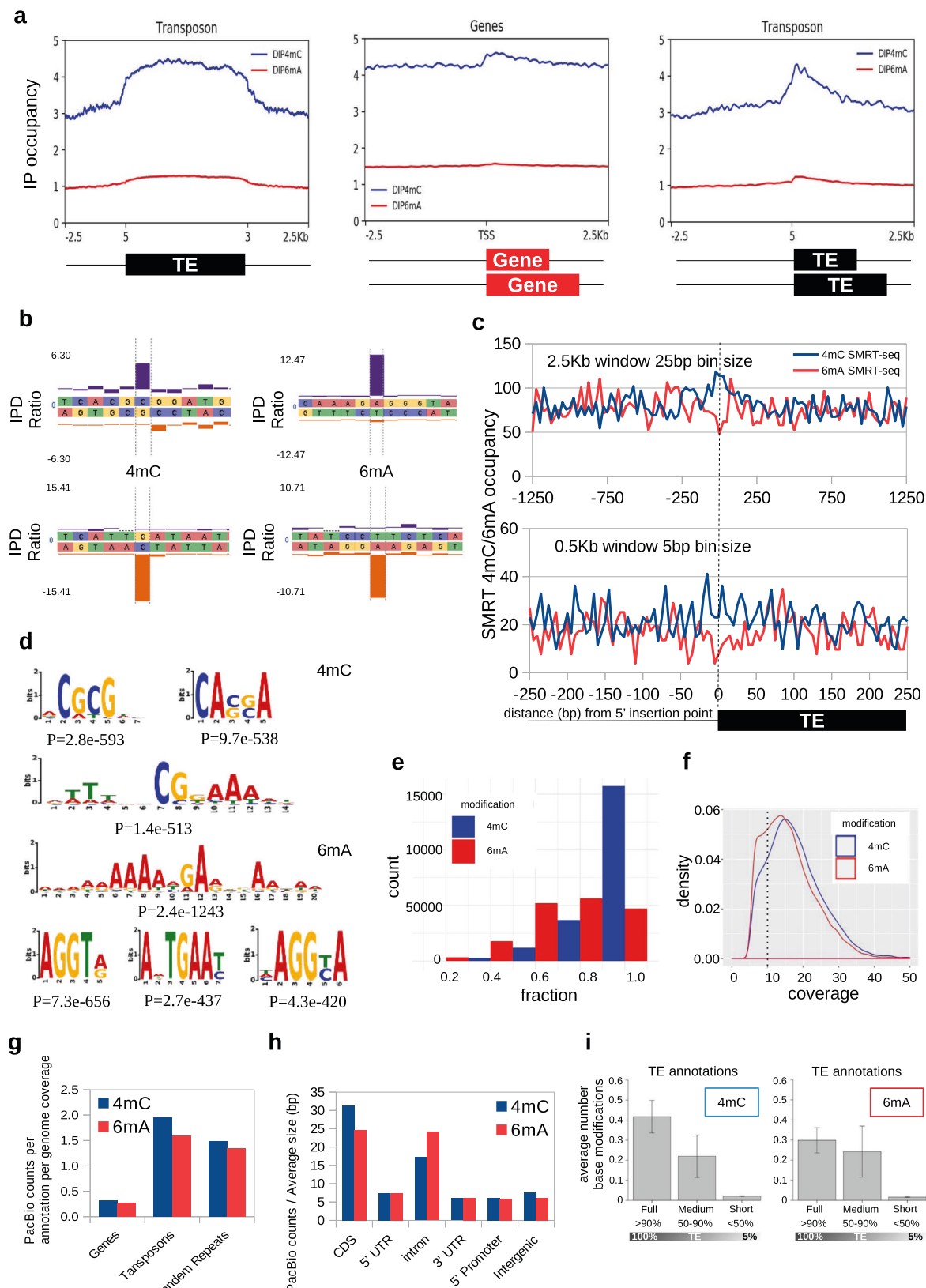

m6A sites in RNA[41], arguing against RNA contamination. The dinucleotide GA is the most prevalent at 6mA sites, and the most common triplets AGG or GAA, when combined, compose 34% of all 6mA triplets. These findings parallel the known 6mA motif preferences in metazoans but differ from unicellular eukaryotes and early diverging fungi, in which 6mA methylation is symmetric and targets ApT dinucleotides (Supplementary Table 1).

In addition to measuring coverage at each 4mC and 6mA site, the SMRT-analysis pipeline reports different methylation levels (fraction), referring to the proportion of times a given nucleotide is identified as methylated (1 equals 100% methylation). Notably,

**Fig. 2 Genome-wide distribution of 4mC and 6mA methylation in *A. vaga*. a** Distribution of DIP-seq 4mC and 6mA sites around TEs (metaprofile), genes (TSS), and 5′-end TE profiles in Av-ref, showing IP occupancy in 25-bp bins within ±2.5 kb of each feature. In metaprofiles, the body size feature, representing genes or TEs, is automated and normalized (0–100% length). TE 5′-profile shows 4mC and 6mA sites near 5′ boundaries, aligning transposons at the 5′ end. **b** IPD ratios in AvL1 SMRT-seq data at four representative 4mC and 6mA modification sites. Purple and orange bars, Watson and Crick strands. **c** SMRT-seq 4mC and 6mA occupancy in 5 and 25-bp bin sizes within ±0.5 and ±2.5 kb of 5′ TE boundaries. **d** MEME-ChIP motif analysis of regions around SMRT-seq 4mC and 6mA sites. Windows of ±5, ±10, and ±20 bp were extracted and searched for significant motif enrichment. Significance was assessed by Fisher's exact test; *p*-value generated by MEME-ChIP is shown under each motif. **e** Methylation fraction distribution at modified sites detected by SMRT-seq. Most 4mC sites are fully methylated (fraction = 1); average methylation level of 6mA sites is 0.74. **f** PacBio read coverage distribution by base modification sites. The minimal threshold coverage limit applied for calling 4mC and 6mA methylated sites to calculate methylation fraction per site in (**e**) is shown by a dashed line. **g** Average numbers of 4mC and 6mA base modifications in protein-coding genes, TEs, and tandem repeats. Average is calculated as the total number of modified sites divided by the total number of annotations (unique IDs) in each feature and divided (normalized) by the genome fraction covered by such annotation in the genome (genes, 0.533; TE, 0.021; TR, 0.0084). **h** Distribution of SMRT-seq 4mC and 6mA sites within genic features (CDS, intron, 5′ UTR, 3′ UTR, 5′-promoter region) and intergenic regions by average feature size (bp). **i** DNA methylation density vs. TE copy integrity. Bar height indicates average number of 4mC or 6mA SMRT counts; error bars represent standard deviation for full (*n* = 321), medium (*n* = 305), and short (*n* = 8623) TE copies.

most 4mC methylation corresponds to high-fraction sites (0.5–1), dominating over low-fraction sites (0.1–0.5) at a ratio 71:1, with 58% of 4mC sites being fully methylated (Fig. 2e). Methylation at 6mA sites appears more dynamic, although the highly methylated (0.8–1) and moderately methylated (0.5–0.8) sites still dominate over low-fraction sites (0.1–0.5), which constitute only 12% of 6mA sites.

We plotted the density of 4mC and 6mA in AvL1 (DIP-seq and SMRT-seq) across annotated features (genes, TEs, tandem repeats (TRs)) (Fig. 2c, g; Supplementary Fig. 5a–d). The 4mC and 6mA tag densities are close for each annotation type (Fig. 2g). The 4mC density in SMRT-seq is higher near TE 5′-ends (Fig. 2c), as was also seen in Av-ref DIP-seq showing increased deposition of 4mC peaks close to TE 5′ ends (Fig. 2a, right). Enrichment of 4mC over TEs, especially near 5′-boundaries, was also preserved for fractional methylation values in SMRT-seq metaprofiles (Supplementary Fig. 6b). Nevertheless, a fair number of 6mA sites (DIP-seq and SMRT-seq) is found near TEs (Fig. 2g, i; Supplementary Fig. 5b, d).

Methylation density in TRs deserves special mention. Figure 2g shows that the average counts of 4mC and 6mA sites in TRs are elevated in comparison with TEs and genes. According to TR annotation (Methods), only a small fraction (0.84%) of the AvL1 assembly is composed of TRs. Inspection of SMRT-seq modification data identified two repeats with a very high density of methylated sites, located mainly on contigs 1882 and 785 adjacent to large *Athena* retroelements[42]. Such extra-high modification density, approaching that in bacterial contigs, mostly accounts for over-representation of modified bases in TRs, leaving other TRs virtually unmethylated. In subsequent experiments, we took advantage of the high methylation susceptibility of these repeats (see below).

In genes, the PacBio methylation tag density is much lower than that in TEs and TRs (Fig. 2g). Still, genic regions cover slightly over one-half of the AvL1 genome, attracting a sizeable proportion of 4mC and 6mA modifications (52% of 4mC and 54% of 6mA). To correlate methyl marks with gene structure, we examined 4mC and 6mA distribution using more refined features: gene bodies, promoters within 2 kb upstream of the TSS, and intergenic regions which may include TEs and TRs, with gene bodies further subdivided into CDS (exons excluding 5′ and 3′ UTRs), introns, 5′ and 3′ UTRs (Fig. 2h). Altogether, base modifications are found in all features (CDS, promoters, and intergenic regions); when the density per average feature size is compared, CDS regions carry more 4mC than introns (Fig. 2h), reminiscent of 5mC patterns in mammals[43], but introns carry as many 6mA marks as CDS, minimizing the possibility of m6A carryover from RNA.

In AvL1, DIP-seq shows relative enrichment with 4mC and 6mA within TE bodies (Supplementary Fig. 5b, d). PacBio 4mC sites display a trend for enrichment near the 5′ TE boundaries, while 6mA sites show a local depletion (Fig. 2c), which is visible even though TE promoters are located near TE 5′-ends but not necessarily at the boundary, and is not due to a local change in base or dinucleotide composition (Supplementary Fig. 3b). Moreover, 4mC and 6mA marks are predominantly found over full-length or nearly full-length TE copies and are practically absent from shorter TE fragments spanning less than one-half of TE consensus length, suggesting that active TE copies are preferentially targeted (Fig. 2i). The lack of 4mC and 6mA marks in shorter TE copies, together with a concentration of 4mC near 5′ TE boundaries, suggest that their deposition is associated with transcriptional activity.

To visualize 4mC and 6mA densities in TRs, TEs, and genes on representative contigs, we built Circos plots (Supplementary Fig. 7a–d), in which the PacBio modification layer is plotted as modification fraction (from 0 to 1) for each modified base. In agreement with Fig. 2e, highly methylated 4mC sites dominate in most locations, while 6mA sites are distributed over a much wider methylated fraction range and across a wider feature range. Importantly, higher densities of modified bases are not necessarily correlated with areas of higher PacBio read coverage, indicating that over-representation of methyl marks over TEs and TRs is not due to excess coverage in these regions (e.g. mtDNA at 127x coverage displays very few marks) (Supplementary Fig. 7e). Supplementary Fig. 7c, d shows that long copies of *Vesta* and *Athena* retrotransposons attract methyl marks, but short copies do not. Supplementary Fig. 8 presents a more detailed view of selected contigs, including TRs, retroelements, and DNA TEs. Of note, an inspection of 36 high-density 4mC regions lacking annotations showed that one-half correspond to TEs unrecognized during annotation, independently confirming TEs as N4CMT targets (Supplementary Fig. 7f; Supplementary Table 11; Supplementary Note 3).

**N4CMT acts as 4mC-methyltransferase in *E. coli*.** The domain structure of N4CMT cannot be taken as evidence of its N4C-MTase activity, since the *N6_N4_MTase* domain repeatedly evolved 6mA or 4mC specificities[44]. However, N4CMT function cannot be disrupted in vivo, as the tools for genetic manipulation in bdelloids are yet to be developed. We, therefore, sought to investigate the activity of the recombinant N4CMT protein in a heterologous system. To this end, we PCR-amplified N4CMT from *A. vaga* cDNA to obtain intronless versions (Methods; Supplementary Table 7). Amplicons were cloned into pET29b expression vector with the N-terminal S-tag and the C-terminal

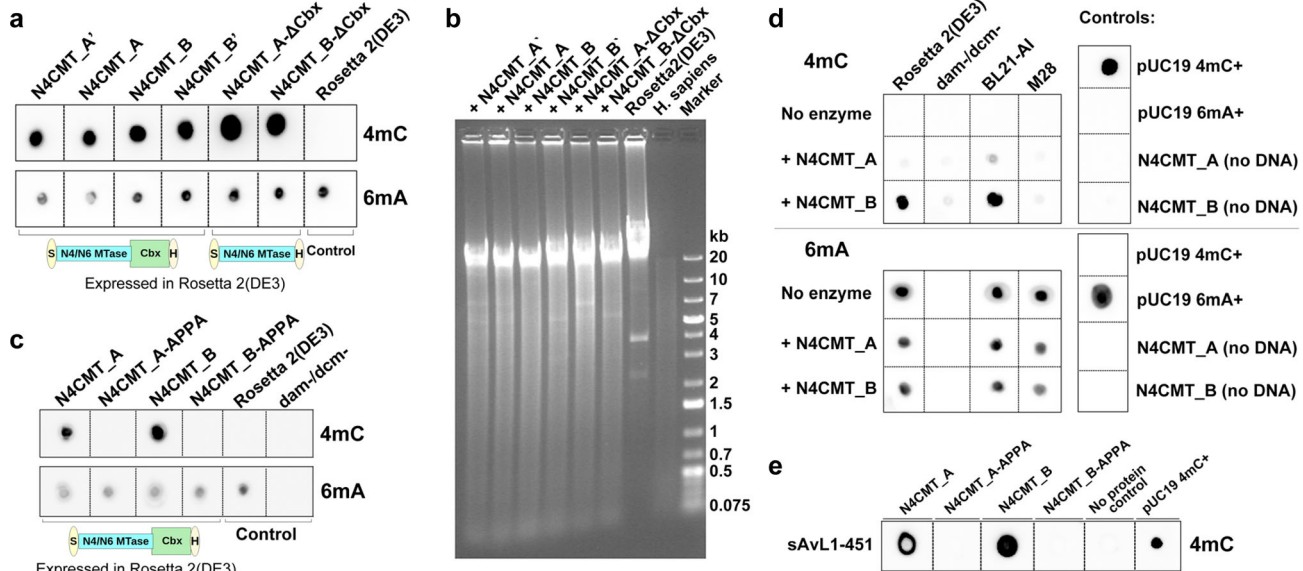

**Fig. 3 N4CMT activity in *E. coli*. a** Immuno-dot blot of total DNA extracted from *E. coli* Rosetta 2(DE3) strain transformed with different variants of recombinant N4CMT. DNA from non-transformed Rosetta 2(DE3) was used as a control. DNA was extracted after 4 h of IPTG-induced N4CMT expression; 400 ng of each DNA was spotted in one dot. Methyl marks are indicated on the right. **b** Methylcytosine-sensitive digestion of *E. coli* gDNA. Total DNA from Rosetta 2(DE3) strain, either transformed with recombinant N4CMTs (as shown on the top) or untransformed, was extracted 4 h post-induction; DNA (6.3 µg) was treated with McrBC. DNA from HepG2 liver cell line (*H. sapiens*) was used as a positive control (500 ng). Two independent experiments were performed with similar results. **c** Immuno-dot blot of total *E. coli* DNA showing the role of the SPPY motif in N4CMT methylation. Same designations as in (**a**). **d**, Immuno-dot blot of total *E. coli* DNA extracted from different strains and treated with N4CMT. Rosetta 2(DE3) and *dam−/dcm−*, 950 ng per dot; BL21-AI and M28, 500 ng per dot. **e** Immuno-dot blot with anti-4mC antibody for sAvL1-451 substrate treated with N4CMT allozymes and their catalytic site mutants in vitro.

6×His-tag and expressed in *E. coli*. We examined two *A. vaga* allozymes A and B, differing by six amino acids (aa): three in the *N6_N4_MTase* domain and three in the chromodomain-containing C-terminus (Supplementary Table 8; Supplementary Fig. 9a). We also tested two inter-allelic recombinants swapping the rightmost substitution near the C-terminal His-tag, which may have arisen during rotifer cultivation or PCR amplification, as well as two 3′-truncated derivatives lacking the chromodomain.

To assess plasmid-borne N4CMT activity in vivo, its expression was induced by IPTG, and gDNA was extracted 4 h post-induction (Methods). Figure 3a shows the immuno-dot-blot of membrane-immobilized gDNAs probed with anti-4mC and anti-6mA antibodies, with 4mC signal observed from full-length N4CMT allozymes in the absence of signal from the untransformed host strain. As expected in the *dam*+ background, 6mA methylation was detected in all samples, serving as an internal DNA control. Not surprisingly, removal of the chromodomain, which yields a core MTase equal in length to its bacterial counterparts, did not reduce activity and even showed an increase in signal intensity due to better solubility of the 33- vs. 45-kDa enzyme (Fig. 3a, N4CMT-ΔCbx). The N4CMT_A allozyme mostly showed weaker activity, suggesting that substitutions in the presumed TRD region of the *N6_N4_MTase* domain affect protein solubility or interaction with target DNA. These findings were corroborated by digestion of corresponding gDNAs with the endonuclease McrBC, which cleaves DNA at any modified cytosines. DNAs extracted from Rosetta 2(DE3) transformed with six N4CMT-expressing plasmids and the control human DNA were readily digested with McrBC, while DNA from the untransformed *dcm*- strain was not (Fig. 3b).

To ensure that the observed activity is directly attributable to N4CMT, we created N4CMT mutants in which the catalytic SPPY motif was replaced with APPA (Supplementary Table 8). Figure 3c shows that 4mC addition is abolished after substitution

of the catalytic Ser and Tyr residues with Ala, indicating that N4CMT is responsible for adding N4-methyl groups to cytosines in dsDNA with SPPY as the catalytic motif, justifying our initial N4CMT designation. Further investigation of purified recombinant N4CMT activity on preferred substrates in vitro revealed that it acts de novo on unmethylated dsDNA, and that a conserved sequence motif in the TR mediates sequence-specific mode of substrate recognition (Supplementary Note 2; Supplementary Fig. 9).

**Base modifications and histone modifications.** In the context of eukaryotic chromosomal DNA environment, any intrinsic target preferences of N4CMT manifested in vitro, while apparently yielding high 4mC densities in certain TRs, would not necessarily be required for 4mC deposition in other genomic regions, which may instead be facilitated by the C-terminal chromodomain of the chromobox type (CBX)[34]. CBX is expected to recognize methylated lysines K9 and K27, the best-studied heterochromatic marks embedded in the ARKS motif at the N-terminus of histone H3, which are associated with transcriptionally silent chromatin and in non-mammalian systems frequently overlap, not because of antibody cross-reactivity but due to a similar function in TE repression[45–48]. To associate DNA methylation marks with specific histone modifications, we performed chromatin immuno-precipitation followed by deep sequencing (ChIP-seq) on *A. vaga* chromatin with anti-H3K9me3 and anti-H3K27me3 antibodies (Methods). For contrasting comparisons with active chromatin, we used an anti-H3K4me3 antibody, which recognizes the modification associated with active TSS[45,49]. After validating antibodies by immuno-dot-blotting (Methods), we profiled the distribution of three H3K modifications in Av-ref and AvL1 strains by ChIP-seq. We found that H3K9me3, a mark for constitutive heterochromatin, often co-localizes with H3K27me3

known to characterize facultative heterochromatin, but not with H3K4me3 which marks active genes (Supplementary Table 10). As expected, host genes display significant H3K4me3 enrichment, which typically covers 1–2 kb around the TSS and shows a characteristic bimodal peak in both strains (Fig. 4a, c top). In contrast, H3K9me3 and H3K27me3 enrichment is observed mostly over TEs and covers the entire TE body, often extending upstream and downstream from a TE insertion, which may be indicative of spreading (Fig. 4b, d top).

To explore the association of 4mC and 6mA with active or repressive histone marks, we used ChIP-seq data for the euchromatic mark (H3K4me3) and two heterochromatic marks (H3K9me3 and H3K27me3) as a proxy for active and silent chromatin, respectively. The low resolution of DIP-seq precludes genome-wide extrapolations in Av-ref, allowing only initial comparisons. For 6mA-DIP-seq peaks, 13.6% intersected with regions bearing euchromatic histone modifications (H3K4me3), while only 4.4% overlapped with heterochromatic histone modifications (H3K9me3 and H3K27me3 combined). For 4mC-DIP-seq peaks, 6.5% intersected with regions bearing heterochromatic histone modifications (H3K9me3 and H3K27me3), but only a minor fraction (1.5%) overlapped with H3K4-marked regions. Following normalization and aggregation of aligned reads in ChIP-seq datasets and comparison with chromatin DNA input (log2 ratio with bamCompare; see "Methods"), we found that DIP-seq peak summits (4mC and 6mA) overlap with H3K9me3 and H3K27me3 ChIP-seq covered regions, while little if any overlap is seen with H3K4me3 (Fig. 4e).

In AvL1, for 4mC-DIP-seq peaks, 42.3% intersected with regions bearing heterochromatic histone modifications (H3K9me3 and H3K27me3 combined), but only 6.6% overlapped with H3K4me3-marked regions. Similarly, for 6mA-DIP-seq peaks, 42.9% overlapped with heterochromatic histone modifications (H3K9me3 and H3K27me3 combined), but only 6.3% intersected with regions bearing euchromatic H3K4me3 modifications. After normalization of aligned reads in the ChIP-seq dataset and comparison with chromatin DNA input, we confirmed that DIP-seq peak summits (4mC and 6mA) are strongly correlated with H3K9me3 and H3K27me3 heterochromatic ChIP-seq reads, as seen in Fig. 4f. Examples of co-localization may be seen in Fig. 4h and Supplementary Fig. 8. Thus, the presence of DNA methyl marks is preferentially associated with silent chromatin in both strains. A similar pattern is observed in AvL1 SMRT analysis, where the 4mC and 6mA marks are more frequently associated with inactive chromatin domains marked by H3K9me3 and especially H3K27me3 (Fig. 4g). Collectively, these results support the view that, in addition to any intrinsic target preferences of N4CMT, its action in the genome may be directed by the CBX moiety, targeting MTase activity to chromatin regions with repressive histone marks.

**Methylomes and transcriptomes in the chromatin context**. To associate histone marks with transcriptionally active or repressed genes in *A. vaga*, we plotted our RNA-seq data for genes co-localizing with either active or repressive H3K-me3 histone marks ("Methods"). As expected, genes near H3K4me3 have significantly higher RPKM (reads per kilobase of transcript per million mapped reads) values (ANOVA $p$-val < 0.01) than genes with heterochromatic histone marks (H3K9me3 and H3K27me3) or no marks (Fig. 5a). AvL1 displays the same pattern (Supplementary Fig. 10a). The tentative designation of 6mA modification as an active epigenetic mark[9,13] prompted us to similarly explore its correlation with gene transcription. The *A. vaga* gene dataset, after removing TE-derived genes, was divided into two groups, with and without the presence of 6mA peaks within a window size of ±500 bp of each gene ID, and RPKM

values were counted in both groups. We found that genes with 6mA depositions tend to have higher RPKM than genes without 6mA ($t$-test $p$-val: 2.2E−16, Fig. 5b bottom). For genes with 4mC modifications, no significant differences in expression were seen with or without 4mC marks (Fig. 5b top). A detailed analysis of 6mA distribution in genes and their promoters, which shows that only a subset of genes is affected, and rules out contribution of m$^6$A from RNA, is presented in Supplementary Note 3 and Supplementary Figs. 11 and 12 (see also Source Data 1).

A different picture was observed for TEs upon examining the association of transcript levels of TE-related genes with DIP-seq peaks ("Methods"). While TE-related genes with or without 6mA did not show much difference in RPKM values, TE-related genes with 4mC marks showed a decrease when compared to those without 4mC ($t$-test $p$-val: 6.8E−8, Fig. 5c, top). Thus, in expressed TEs 4mC may be regarded as a repressive mark. Note that co-localization of 4mC and 6mA is compatible with repression, as 6mA was reported to form an adversarial network preserving Polycomb silencing[23]. Alternatively, some of the 6mA marks co-localizing with 4mC-rich regions may represent a "bleed-through" signal from the nearby 4mC in SMRT-seq data, as was inferred for 5mC-rich regions in mammals[11]. Regardless of 6mA role, the transcriptionally repressed state of TEs is corroborated by a measurable overlap with small RNA profiles, observed for 4mC but not for 6mA (Supplementary Note 4; Supplementary Fig. 13c). Small RNAs play a prominent role in transcriptional repression of *A. vaga* TEs[50], and bdelloids show a dramatic expansion of RNA-mediated silencing machinery, with dozens of Piwi/Ago and RdRP copies[17,51].

**Interpreting the 4mC marks**. To identify possible readers of bacterial marks, we searched for candidate proteins capable of discriminating between methylated and unmethylated cytosines. All known DNA methyl groups protrude from the major groove of the B-form double helix and can be recognized as epigenetic marks. In eukaryotes, several protein domains can read 5mC (SRA/SAD/YDG; MBD/TAM; Kaiso) or 6mA (HARE-HTH; RAMA)[6,12], usually in a preferred sequence context. We used profile-HMM searches to find candidate methyl readers in *Adineta* genomes. No homologs were found for the SAD_SRA domain (PF02182), which recognizes hemimethylated CpGs by embracing DNA and flipping out the methylated cytosine[52]. However, we saw the drastic expansion of MBD/TAM-containing proteins, which do not require base-flipping: 14 different alleles (originating from three quartets, Q1–Q3, plus a segmental duplication) encode seven SETDB1 variants, as opposed to only one in monogonont rotifers or other invertebrates (Fig. 6a, b; Supplementary Fig. 14a; Supplementary Data 2). These proteins share the same domain architecture, with the MBD sandwiched between the N-terminal triple-Tudor domains and the C-terminal pre-SET/SET/post-SET domains, present in all SETDB1/eggless-like H3K9me3 histone lysine MTases (KMTs) (Fig. 6a). All seven proteins are transcribed in each *Adineta spp.* (Supplementary Fig. 15). Additional MBD/TAM domains of BAZ2A/TIP5-like remodelers, which form heterochromatin on rDNA and satellites[53], comprise only one quartet in *A. vaga* (Supplementary Fig. 14c; Supplementary Data 3).

To find out whether other KMTs are similarly expanded, we performed an inventory of SET domain-containing *A. vaga* proteins, especially those known to methylate H3K9/H3K27 (Supplementary Data 3). In addition to seven pairs of SETDB1 homologs acting on H3K9, we detected two quartets of E(z)/EZH/mes-2-like orthologs (KOG1079, Transcriptional repressor Ezh1) known to methylate H3K27. More distantly related SET-domain proteins showed domain architectures characteristic of H3K4,

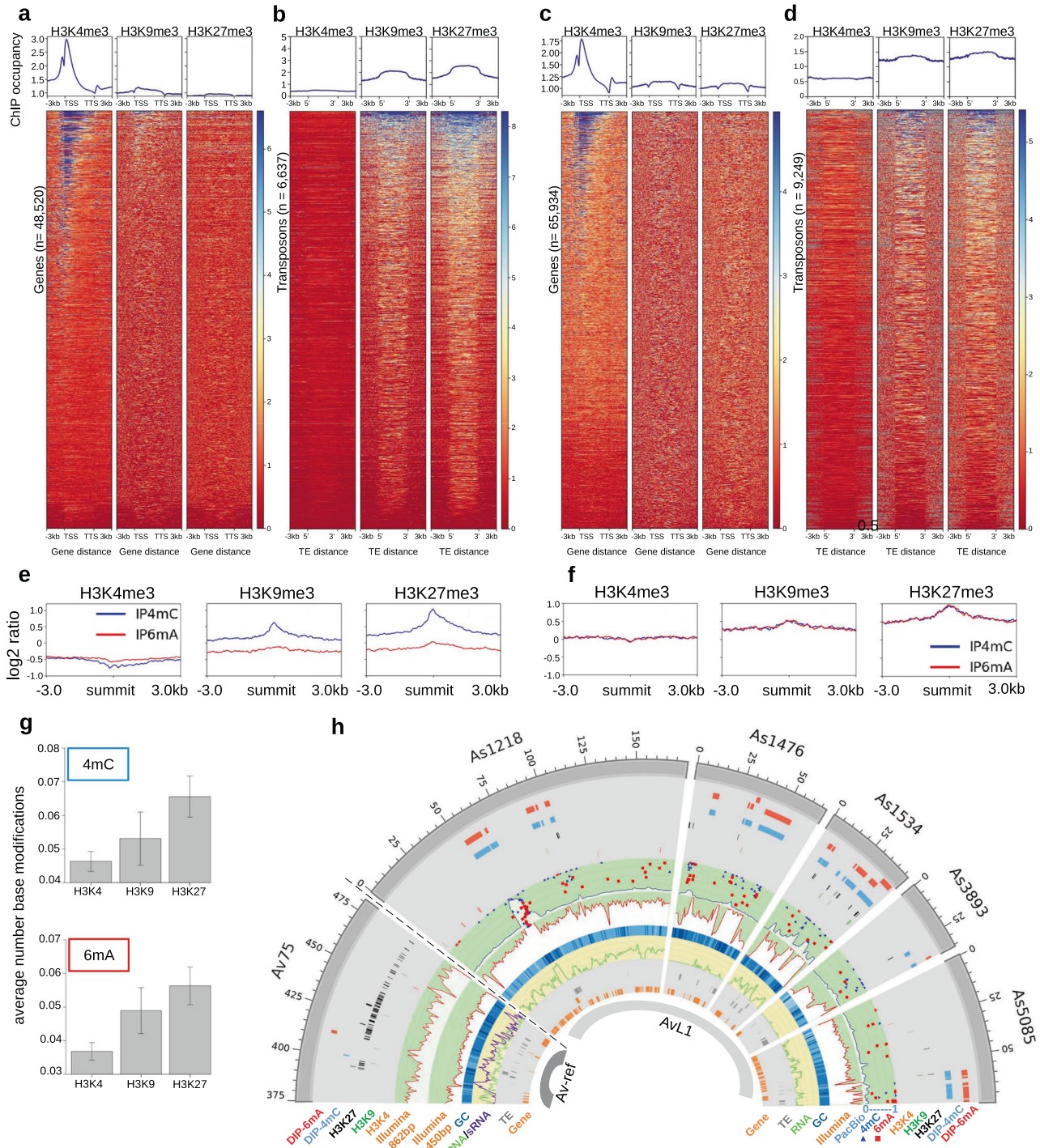

**Fig. 4 Base modifications and histone modifications.** Shown are the *A. vaga* strains Av-ref (**a**, **b**, **e**) and AvL1 (**c**, **d**, **f**). **a**, **c** Profiles and heatmaps for gene regions with transcription start sites (TSS) and transcription termination sites (TTS). Profiles (top) show mean ChIP-seq signal for H3K4me3, H3K9me3, and H3K27me3. Heatmaps (bottom) represent H3K4me3, H3K9me3, and H3K27me3 ChIP-seq reads, where each row corresponds to gene regions with ±3 kb from TSS and TTS boundaries. Heatmap color bars represent H3K enrichment normalized by RPGC (reads per genome coverage). **b**, **d** Profiles and heatmaps for TE annotations delimited by 5′ and 3′ boundaries. Profiles (top) show mean ChIP-seq signal for H3K4me3, H3K9me3, and H3K27me3. Heatmaps (bottom) represent H3K4me3, H3K9me3, and H3K27me3 ChIP-seq reads, where each row corresponds to ±3 kb from 5′ or 3′ TE boundary. **e**, **f** Intersection of 4mC and 6mA DIP-seq peak summits with ChIP-seq for histone modification marks H3K4me3, H3K9me3, and H3K27me3 for Av-ref (**e**) and AvL1 (**f**). The log2 ratio is shown over a scaled window ±3 kb. *y*-Axis, relative fold enrichment. **g** DNA base methylation counts near H3K4me3, H3K9me3, and H3K27me3 ChIP-seq peaks. Counts are taken around each peak in a ±500 bp window. The bar height shows an average number of counts (SMRT-seq 4mC and 6mA). Error bars represent standard deviation for H3K4 (*n* = 5789), H3K9 (*n* = 1205), and H3K27 (*n* = 2378) peaks. **h** Circos plot illustrating DIP/ChIP peaks, methylation sites, sequencing read coverage, and gene/TE annotations in selected Av-ref and AvL1 contigs. Features are explained in the key; further details are in Methods and Supplementary Fig. 7. SMRT-seq DNA methylation marks are shown within the PacBio layer in AvL1 for 4mC (blue triangle) and 6mA (red square). Mark height in the ring shows a methylation fraction (0–1). Green line, RNA-seq coverage; purple line, small RNA coverage in Av-ref.

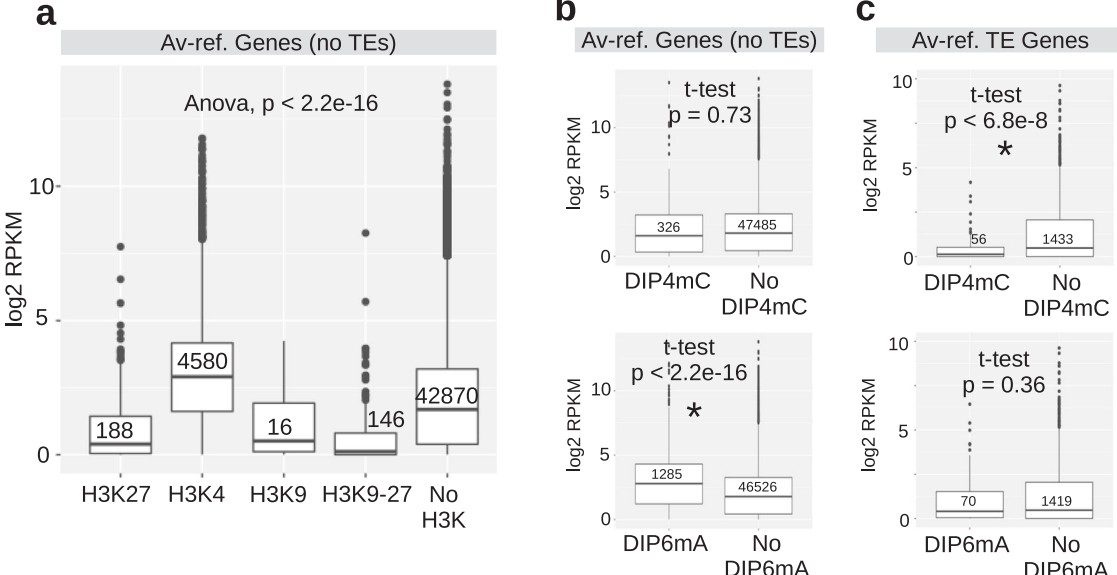

**Fig. 5 Association of transcript levels with histone and DNA methylation. a** Box plot showing Av-ref gene expression levels (log$_2$RPKM) associated with co-localized H3K4me3, H3K9me3, H3K27me3, and both H3K9-27me3 marks or without histone marks. ANOVA analysis (one-sided with Tukey's post hoc; $p$-val < 2.2E−16) shows significant differences in expression, with genes associated with the H3K4me3 mark displaying the highest RPKM (reads per kilobase per million mapped reads). **b**, **c** RPKM values associated with DIP-seq 4mC and 6mA base modifications in *A. vaga*. TE genes in (**c**) are derived from Av-ref automated gene models after positively intersecting with TE annotations. Box plots indicate the first and third quartiles; line, the median; single points as outliers. Error bars are calculated as the standard deviation of three biological replicates. The p-values were calculated by a two-tailed Student's *t*-test, with asterisks indicating significant differences.

H3K36, and H4K20 KMTs (Trx-G/Ash1/Set1/MLL, SETD2, SETD8) and were not expanded, comprising either a quartet or a pair. Interestingly, we found six stand-alone SET-domain homology regions resembling H3K4/H3K36 KMTs (PRDM9/7/set-17), which were not predicted in the annotated gene set, non-transcribed, and lacked additional domains (KRAB_A-box, SSXRD) characteristic of PRDM9/7 proteins involved in localizing meiotic recombination hotspots and in male-specific expression[54,55]. Unexpectedly, we failed to identify two known KMT types acting on H3K9 or K9/K27: Su(var)3-9/SUV39H1/set-25/Clr4, a "histone read-write" architecture consisting of chromo- and SET-domains, mediating constitutive heterochromatin formation[56]; and G9a/EHMT2/KMT1C (ankyrin repeats plus SET), which initiates de novo methylation and silencing of repeats and developmentally regulated genes[57]. These domain architectures may have been lost and/or replaced by vastly expanded SETDB1-like variants.

We next sought to determine whether SETDB1 is similarly amplified in all bdelloids. Six species in the genus *Rotaria* from the family Philodinidae[19] possess the same seven variants as do Adinetidae, indicating that SETDB1 amplification occurred prior to divergence of the major bdelloid families (Fig. 6b). An unusual SETDB1 divergence pattern is seen in the bdelloid *Didymodactylos carnosus*, which forms the deepest-branching sister clade to other known bdelloids[51] and lacks N4CMT. While in three cases Dcar_SETDB1 forms sister clades to variants from other bdelloids, preceding quartet formation, the Q1 quartet lacks Dcar_SETDB1 homologs; moreover, an ortholog of Av_s314 shows clear evidence of loss, detected as a small 170-aa C-terminal fragment (Supplementary Data 3). This natural gene knockout is associated with an increase in LINE elements to the levels seen in monogononts, which agrees well with high concentration of 4mC over LINEs (Supplementary Fig. 16), but was not prevented by high copy number of Ago/Piwi proteins (Fig. 6c)[51]. Notably, LINE elements, due to their mostly vertical transmission, are expected to be more deleterious if sex is rare or absent[58].

The role of MBD as a universal discriminator of 5mC marks in DNA is questioned by the presence of SETDB1 orthologs in species lacking 5mC, such as *Drosophila melanogaster* and *Caenorhabditis elegans*[6], and many MBD proteins do not bind 5mC (Supplementary Fig. 14c). However, the structure of human MBD1 shows its unique potential for recognizing 5mC in the major groove without encircling DNA, which makes MBD an ideal candidate for interacting with nucleosome-bound DNA without interference from core histones[59,60]. Moreover, three of the seven bdelloid SETDB1-like variants display two conserved arginines in the MBD involved in the recognition of cytosines in the DNA backbone, potentially accounting for CpG preference (Supplementary Fig. 14a, b). However, they show extensive variation in the length of the antiparallel β1-β2 loop, which reaches across the major groove and interacts with one of the methyl groups. Since the overall structure is compatible with recognition of an asymmetrical DNA methyl group in the nucleosomal context, we sought to find out whether some of the seven SETDB1 variants may have adapted to preferentially recognize a novel methyl mark in the major groove.

To this end, we synthesized seven recombinant plasmids carrying tagged versions of the corresponding MBD/TAM domains (Supplementary Fig. 17a). We tested these proteins in electrophoretic mobility shift assays (EMSA) with the 451-bp repeat fragment (sAvL1-451, see above), which was either unmethylated or 4mC-methylated by N4CMT in vitro, to ensure sufficient methylation density and favorable position of methyl marks. As MBD/TAM is a generic DNA-binding domain, most AvMBD's are capable of binding both unmethylated and methylated DNA fragments (Supplementary Fig. 17b, c). We chose AvMBD_s314 to assess its binding preference for 4mC-methylated DNA since the loss of its ortholog in *D. carnosus* (see above) is associated with a notable increase in LINE retrotransposon content[51]. We tested four AvMBD_s314 concentrations (2.38, 3.23, 3.75, and 4.14 nM) in EMSA with $^{32}$P-labeled sAvL1-451 and four concentrations of the unlabeled sAvL1-451 competitor, which was either unmethylated or

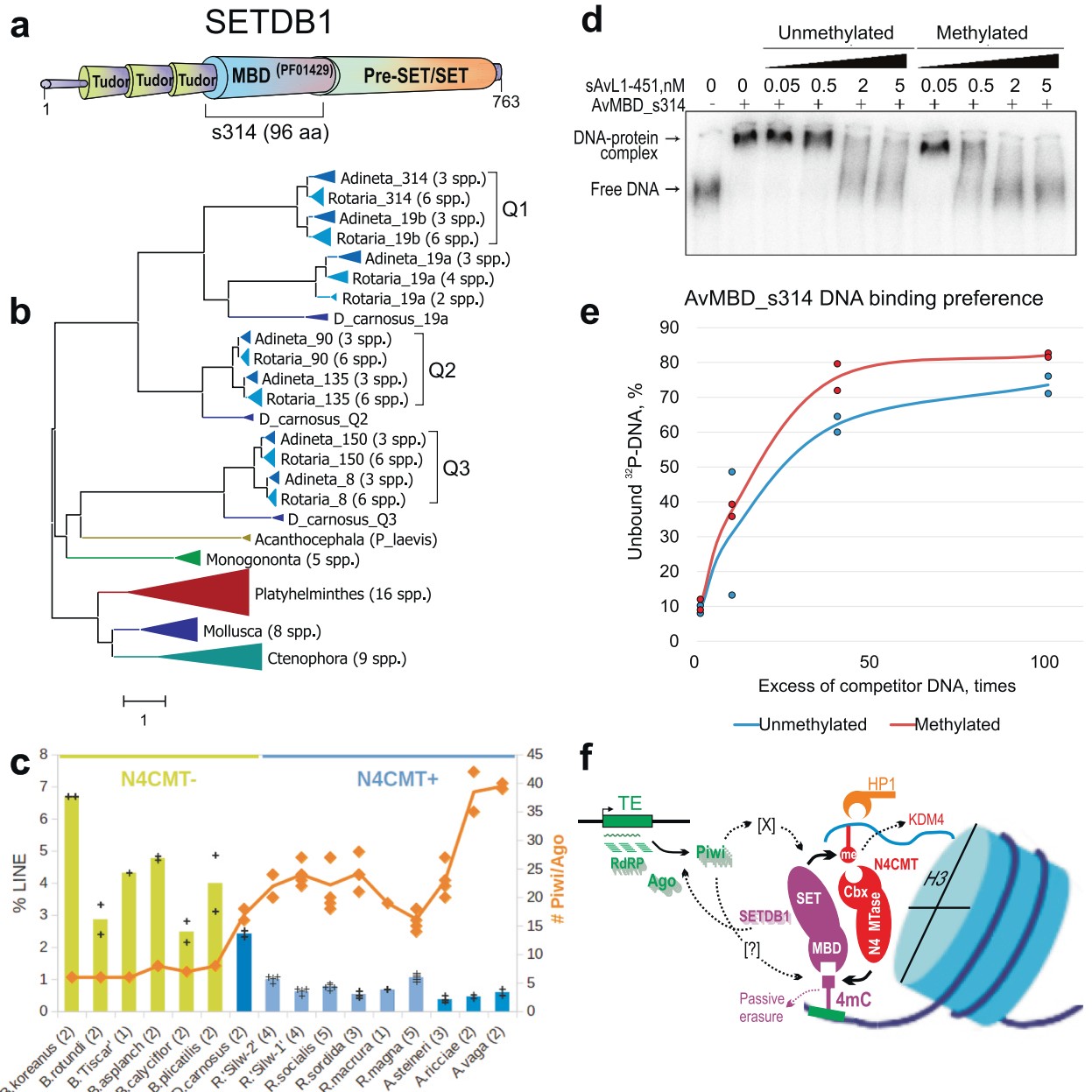

**Fig. 6 Amplification of SETDB1 histone methyltransferases and preference for 4mC-methylated DNA. a** Domain architecture of bdelloid SETDB1 proteins. Square bracket marks cloned MBD domains; aa numbering is for Av_s314. **b** Unrooted maximum likelihood phylogram of SETDB1 variants in bdelloids (blue), monogononts (green), and acanthocephalan (olive). Q1–Q3, quartets of homeologs formed by paleotetraploidy. Bottom clades include single-copy SETDB1 in 3 protostome phyla. See Supplementary Data 2 for aa sequences. Scale bar, aa substitutions per site. **c** LINE retrotransposon content (% genome) and *Piwi/Ago* copy numbers in 6 monogonont (green) and 10 bdelloid (blue) species (Supplementary Table 2; Source Data 2). Median values are indicated by bar heights for LINE content, and by continuous orange line for copy numbers. Data points for % LINE values (+) and *Piwi/Ago* copy numbers (◆) are shown for all sequenced isolates from each species, with number of isolates (*n*) in parentheses. **d** Affinity of AvMBD for 4mC-methylated DNA in electrophoretic mobility shift assays. EMSA was performed using 0.05 nM $^{32}$P-labeled sAvL1-451 DNA and 3.75 nM AvMBD_s314 protein. Unmethylated and 4mC-methylated by N4CMT_B sAvL1-451 fragments were used as competitor DNA. One of two gels with the same DNA:protein ratio from independent experiments is shown. **e** AvMBD_s314 DNA binding preference for methylated DNA. Y-axis, percent unbound $^{32}$P-labeled sAvL1-451 DNA for 3.75 nM AvMBD_s314 in the presence of unmethylated and 4mC-methylated sAvL1-451 competitor DNA. Median values are indicated by continuous lines, with dots representing the data points for two independent experiments. Five independent experiments, including those with different DNA:protein ratios, are presented in Source Data 2. **f** A hypothetical model of the self-reinforcing regulatory loop based on the ability of N4CMT and SETDB1 to cross-recognize methyl marks on histones (circle) and DNA (square), respectively. Shown are the relevant conserved proteins/domains described in the text. Shadows, multiple copies. Hypothesized links with piRNAs/ Piwi (dashed lines) are not defined. [X] and [?] are hypothetical mediator complexes with poorly conserved components, of which only Nxt1 is identifiable in *A. vaga* (see "Discussion"). HP1 and KDM4 are well-conserved. Components involved in other types of histone/DNA modifications are not shown for simplicity.

4mC-methylated by N4CMT_B in vitro. This approach provides a more adequate comparison than measurement of dissociation constants ($K_d$) for two labeled probes, as in vitro methylation is variably efficient. We observed a clear preference of s314 for binding 4mC-methylated DNA, with $p < 0.05$ in a one-tailed Student's $t$-test in four independent experiments (Source Data 2), when using >10× excess of non-labeled competing methylated or unmethylated DNA ($p = 0.044$ for 40×; $p = 0.018$ for 100×). Figure 6d shows a representative EMSA gel for the 3.75 nM s314 protein concentration, which yielded 88.3% protein-bound DNA with 0.05 nM $^{32}$P-labeled sAvL1-451 fragment. This protein concentration was tested twice, and the average change in the amount of unbound DNA over increasing concentrations of unlabeled competitor DNA (Fig. 6e) shows that upon the increase of competitor concentration, the shift from DNA–protein complex to unbound DNA occurs faster for 4mC-modified DNA than for unmethylated DNA, indicating a preference for 4mC target.

## Discussion

Here, we identify and characterize the N4-mC base modification in rotifer DNA, expanding the repertoire of methylated bases in Metazoa with a modification known so far only in bacteria. We confirm its presence in bdelloid rotifers, combining multiple lines of investigation and accounting for artifacts inherent to each modification detection method[38,61,62] and for bacterial contaminations. In agreement with the absence of Dnmt1/Dnmt3-like MTases, we failed to detect 5mC in bdelloids, while 4mC and 6mA were detectable by several orthogonal methods. We identified N4CMT, a horizontally transferred enzyme of bacterial origin, as responsible for the addition of 4mC marks to DNA. Expression of recombinant N4CMT in E. coli results in 4mC addition to DNA, as follows from immuno-dot-blot analysis and methyl-sensitive digests of DNA from N4CMT-expressing bacteria vs. methyl-free strains. Not surprisingly, chromodomain is not required for 4mC deposition either onto bacterial DNA in vivo, which is not packaged into chromatin, or onto preferred DNA substrates by recombinant N4CMT in vitro. However, in the context of eukaryotic chromatin, ChIP-seq and DIP-seq distributions reveal strong correlations between H3K9/27me3 silent chromatin marks and DNA methyl marks. Thus, N4CMT may contribute to epigenetic homeostasis, whereby deposition of repressive chromatin marks is ensured by passive preservation of 4mC via covalent linkage to DNA in the absence of active enzymatic demethylation, helping to maintain TE repression in eggs and adults. Over-representation of 4mC at the 5′ TE boundaries, i.e., near TE promoter regions, may affect transcription factor binding near promoters and cause transcriptional interference, as previously seen for 5mC[63].

While the lack of candidate 4mC erasers supports 4mC role in maintaining TE silencing, other important components of epigenetic systems are the "reader" proteins, which could interpret the 4mC mark to form a regulatory loop, as is known for 5mC and 6mA[64]. The N4CMT architecture is reminiscent of plant chromomethylases (CMT), "histone-read-DNA-write" enzymes with a C5-MTase-embedded chromodomain, which reads H3K9me marks and deposits similar marks at nearby non-CG's. Together with "DNA-read-histone-write" architecture provided by KYP, an H3K9-KMT with the 5mC SRA reader domain, the CMT3-KYP pair forms a mutually reinforcing loop reading each other's epigenetic marks[64]. The crosstalk between mCpG and H3K9me in animals and plants is even more complex, requiring multiple protein factors[5]. In bdelloid N4CMT, a very simple "histone-read-DNA-write" architecture, with the chromodomain reading the repressive H3K9/27me3 marks and MTase writing the atypical 4mC marks onto DNA in the absence of an eraser,

has the potential to link histone and DNA layers through a reinforcement loop which feeds back onto silent chromatin via "DNA-read-histone-write" SETDB1-like KMTs to help maintain repressive marks on histone tails throughout cell divisions for continuous TE silencing (Fig. 6f). Association of 4mC with full-length TEs capable of transcription and the overlap of 4mC and small RNA distribution patterns further suggest that the loop may be triggered by pi-like RNAs from transcribed TEs, which are known to initiate transcriptional silencing on nascent RNAs via Piwi and perhaps SFiNX-like protein complexes[65–67], or may directly affect methylation, as in mice[68]. In this scenario, epigenetic inheritance relies on overriding the normally occurring H3K9me erasure by KDM4/JMJD2[69], which is present in A. vaga. Our finding that an amplified bdelloid-specific SETDB1-like variant prefers binding to 4mC-methylated DNA in vitro suggests that 4mC stimulates more efficient binding of SETDB1 in the nucleosomal context, linking N4CMT-mediated 4mC deposition to re-establishment of H3K9me3 that helps to preserve silent chromatin marks on TEs and other repeats.

Notably, bdelloids exhibit some of the lowest TE content among metazoans, while members of the sister class Monogononta, which lack cytosine methylation and encode a single-copy SETDB1, show reduced ability to contain TE proliferation, which can double their genome size[70]. Earlier, we reported drastic expansion of Ago/Piwi and RdRP genes in bdelloids, which are extremely TE-poor, in contrast to the acanthocephalan Pomphorhynchus laevis (Rotifera) with 66% TE content and no expansion of Ago/Piwi[17,51], underscoring the importance of RNA silencing pathways in TE control. Notably, the bdelloid D. carnosus, despite Ago/Piwi expansion, does not show the dearth of retrotransposons typical of other bdelloids, displaying an elevated content of LINE elements matching that of Brachionus monogononts and shifting the average bdelloid LINE content upwards[51]. Here, we find that D. carnosus lacks N4CMT and specific SETDB1 variants which may have evolved to interact with the 4mC mark, suggesting that the genome defense system in D. carnosus is missing an important layer for preventing TE expansion. The elevated LINE content in this natural knockout of the 4mC-preferring variant highlights the importance of cross-talk between genome defense layers for efficient TE control, since Adineta and Rotaria during their evolutionary history experienced a strong decrease in retrotransposon content (Fig. S3c, h, i in ref. [51]), which coincided with the emergence of N4CMT and of 4mC-preferring SETDB1 variants.

Collectively, our findings help to unravel a fascinating evolutionary puzzle: How can a bacterial enzyme decorating DNA with non-metazoan modifications penetrate eukaryotic gene silencing systems and become preserved by natural selection for tens of millions of years? Given the importance of similar processes at the dawn of eukaryotic evolution, when MTases were recruited to create the extent epigenetic systems, the bdelloid case spans a unique time interval in the evolutionary history, when its advantages have been fully manifested and validated by natural selection, but its resemblance to bacterial counterparts has not yet been completely erased. Losses of DNA methylation have occurred multiple times throughout the eukaryotic tree of life; however, de novo recruitment of a bacterial mark into an existing epigenetic system has not been observed in more recent metazoan history. A synthetic "DNA read-write" 6mA system in cultured human cells, based on E. coli Dam MTase and bypassing chromatin states through artificial targeting, has been created[71], however, such a "shortcut" is unlikely to persist over evolutionary time scales. In the liverwort Marchantia polymorpha, a recently duplicated 4mC methyltransferase of bacterial origin was recruited in spermiogenesis to modify over one-half of all CpG sites, however without additional N- or C-terminal domains it acts genome-wide,

without recognition of specific features[72]. Our system helps to discern the selectively advantageous features in epigenetic control systems and emphasizes that the addition of a DNA epigenetic layer to the histone layer demands enhanced inter-connection of components between layers via acquisition of extra domains for efficient operation. Finally, it demonstrates that horizontal gene transfer, the role of which in eukaryotic regulatory evolution is a subject of intense debate[73,74], can re-shape complex regulatory circuits in metazoans, thereby driving major evolutionary innovations that include epigenetic control systems.

Additional discussion can be found in Supplementary Discussion.

## Methods

**Rotifer cultures**. A clonal culture of *A. vaga*, started in 1995 from a single individual, was maintained continuously in filtered spring water and fed with *E. coli* M28. Rotifers were grown in 150 × 20 mm untreated Petri dishes and transferred into new ones until the desired biomass was reached. The *A. vaga* L1 natural isolate[35] was collected in 2012, and the clonal culture was maintained in the laboratory under the same conditions.

**Plasmid construction**. N4CMT ORFs from scaffold_23 (GSADVT00006927001, allele *N4CMT_A*) and scaffold_179 (GSADVT00035445001 allele *N4CMT_B*) (http://www.genoscope.cns.fr/adineta/cgi-bin/gbrowse/adineta/) were amplified from cDNA to eliminate introns. The first exon in the annotation is variable in different bdelloids, thus it was omitted from primer design, so that the N-terminus coincides with that used by bacterial MTases. Briefly, RNA was extracted from adult rotifers starved for 24 h, using Direct-zol™ RNA Miniprep kit (Zymo Research), and cDNA was synthesized from 2 μg of RNA with SuperScript® IV Reverse Transcriptase (Invitrogen) and random hexamers, following the manufacturer's protocols. *N4CMT* was then amplified by PCR from 5% of cDNA reaction with Q5® Hot Start High-Fidelity DNA Polymerase (NEB). All primers used in this study are listed in Supplementary Table 7. PCR fragments were cloned into pET29b(+) vector (Novagen) using *Bam*HI and *Xho*I sites and were propagated in *E.coli* NEB5-alpha (NEB). Catalytically inactive mutants were obtained using Gen-Edit™ site-directed DNA mutagenesis kit (First Biotech). To obtain substrate plasmids pUC19-m97 and pUC19-m119, the insert sequence was amplified from AvL1 genomic DNA with primers A11motif-Hind3-F and A11motif-BamH1-R (Supplementary Table 7) and OneTaq® Hot Start DNA Polymerase (NEB). Amplicons were treated with *Hin*dIII (Anza™ 16) and *Bam*HI (Anza™ 5) in 1× Anza™ Red Buffer (Thermo Fisher Scientific) and purified through 1.5% agarose gel using Zymoclean Gel DNA Recovery kit (Zymo Research). The pUC19 vector was prepared in the same way, ligated with insert using Instant Sticky-end Ligase Master Mix (NEB), and transformed into NEB5α competent cells (NEB). Plasmid purifications were done with Zyppy Plasmid Miniprep (Zymo Research). Inserts were verified by Sanger sequencing on the ABI3730XL at the W.M. Keck Ecological and Evolutionary Genetics Facility at the Marine Biological Laboratory. Expression plasmids carrying AvMBD inserts in pET29b(+) vector were synthesized by GenScript. All DNA sequences were optimized with Gen-Smart™ service to yield soluble recombinant proteins in *E. coli*.

**Protein expression and purification**. Recombinant proteins were expressed in *E. coli* Rosetta 2(DE3) (Novagen) in LB medium, Miller formulation (Amresco) supplied with 50 μg/ml kanamycin (Fisher Scientific), and 34 μg/ml chloramphenicol (Acros Organic). First, cells were grown at 37 °C, 200 rpm until OD = 0.4. After that, cultures were heat-shocked as follows: 10 min at 42 °C, 20 min at 37 °C, 30 min on ice, and 20 min at 37 °C. After the final OD check, expression of recombinant proteins was induced by supplying the growth medium with IPTG (Gold Bio) to 500 μM, and the culture was grown for an additional 4 h at 32 °C, 350 rpm for N4CMT versions or for an additional 3 h at 34 °C, 300 rpm for AvMBD's. Bacteria were pelleted by centrifugation at 4 °C, 4000*g* for 30 min and stored at −80 °C. Induction of recombinant proteins was confirmed by sodium dodecyl sulphate-polyacrylamide gel electrophoresis (SDS-PAGE) followed by Western blotting as we described in ref. [75]. For protein purification, cellular lysates were prepared using xTractor™ Buffer (Clontech), supplemented with lysozyme (Sigma), DNase I (Promega) or Benzonase® Nuclease (Sigma), and Roche cOmplete™ EDTA-free Protease Inhibitor Cocktail (Sigma), according to the manufacturers' instructions. Soluble proteins were separated from insoluble debris by centrifugation at 4 °C, 4000*g* for 30 min. Recombinant N4CMT were purified using TALON® Single Step Columns (Clontech), following the manufacturer's protocol. Proteins were concentrated using Pierce™ 9 K MWCO Protein Concentrators (Thermo Scientific), and the buffer was exchanged to 50 mM phosphate buffer, 300 mM NaCl, pH 7.0, supplemented with Roche cOmplete™ EDTA-free Protease Inhibitor Cocktail (Sigma). Protein concentrations were equalized based on concentration of the full-length His-tagged protein, as detected by Western blotting with His-tag-specific antibodies (Aviva Systems Biology OAEA00010), using Image Studio™ Lite 5.2.5 software (LI-COR). Purified proteins were stored at 4 °C for up to

2 weeks. Recombinant AvMBD's were purified on ÄKTA Pure M2 with HiTrap TALON 1 ml columns (Cytiva), concentrated with Pierce™ 3 K MWCO Protein Concentrators PES (Thermo Scientific), supplied with EDTA, glycerol, and protease inhibitors to the final buffer composition of 40 mM sodium phosphate, pH 7.4; 240 mM NaCl; 102 mM imidazole; 20% glycerol; 4 mM EDTA; 1× cOmplete protease inhibitor cocktail; 1× Halt protease inhibitor cocktail; pH 7.4. Proteins were stored in single-use aliquots at −80 °C. Proteins were quantified using Micro BCA™ Protein Assay Kit (Thermo Scientific), and their purities were verified by SDS-PAGE in 15% resolving gel followed by staining with InstantBlue Protein Stain (Expedeon) and Western blotting with S-tag (Sigma-Aldrich 71549-3) and His-tag (Aviva Systems Biology OAEA00010) specific antibodies, both at 1:5000 dilutions, as we described in ref. [75].

**DNA substrate preparation for methylation assays**. The *A. vaga* cultures were maintained as above but fed with *dam−/dcm− E. coli* (C2925, NEB) strain instead of M28 for a month. Genomic DNA was extracted from adult rotifers starved for 48 h, following the standard phenol–chloroform extraction protocol[76]. To obtain control DNA from different *E. coli* strains (Supplementary Table 3), bacteria were grown overnight in LB medium Miller formulation (Amresco) at 37 °C and 200 rpm, and total DNA was extracted using UltraClean® Microbial DNA Isolation Kit (MoBio Labs).

For N4CMT in vivo activity assays, plasmids carrying *N4CMT* inserts were transformed into Rosetta 2(DE3) strain. Bacteria were grown as above, pelleted and stored at −80 °C until expression of recombinant proteins was confirmed by Western blot hybridization with His-tag-specific antibodies. After that, bacterial pellets were incubated in lysis buffer (10 mM Tris, pH 8.0, 100 mM NaCl, 5 mM EDTA, 120 μg/ml Proteinase K (ThermoFischer), 0.6% SDS) at 53 °C overnight. Total DNA was purified using the standard phenol-chloroform extraction protocol[76], including treatment with RNaseONE (Promega). DNA quantity and quality were inspected by agarose gel electrophoresis and NanoDrop 2.0 measurements. Cleavage of gDNA by McrBC (NEB) was performed overnight at 37 °C as recommended by the manufacturer, followed by separation in 0.8% TAE-agarose gel electrophoresis. Plasmids (pUC19, pBlueScript SK+, etc.) for methyltransferase assays were transformed into methyl-free C2925 competent cells (NEB) and purified using Zyppy Plasmid Miniprep (Zymo Research). To obtain 4mC-positive control for immunoassays, pUC19 was methylated with *M.Bam*HI MTase (NEB). To obtain a positive control for 6mA, pUC19 was purified from NEB5α (*dam*+) *E. coli* strain. Oligonucleotides were ordered from Eurofins Genomics and annealed in 1× annealing buffer (10 mM Tris, pH 7.5, 50 mM NaCl, 1 mM EDTA) as follows: the mix was incubated at 95 °C for 3 min and allowed to cool down to RT for 1 h. Other dsDNA substrates were obtained by PCR and purified using Monarch PCR clean-up kit (NEB) or Zymoclean Gel DNA Recovery kit (Zymo Research).

**In vitro methyltransferase activity assays**. Reactions were carried in 1× *M.Bam*HI Methyltransferase Reaction Buffer (NEB) supplemented with 80 μM *S*-adenosyl-L-methionine (SAM) provided with the buffer. Optimal results were obtained with 500 μg/ml as a final concentration of N4CMT recombinant proteins. Reactions were initially incubated at 25 °C for 4 h, and incubation was continued for another 16 h after supplementing with additional 80 μM SAM.

**DNA dot blot immunoassays**. Samples were spotted on BioTrace™ NT Nitro-cellulose Transfer Membrane (Pall Corporation), air-dried and UV-cross-linked with 120,000 μJ/cm² exposure using Spectrolinker™ XL-1500 UV crosslinker (Spectronics Corporation). The cross-linked membrane was blocked in 3% non-fat milk in TBST (containing 0.05% v/v Tween) and incubated with 1:40,000 anti-N4-methyl-C antibody or with 1:60,000 anti-N6-methyl-A antibody at 25 °C for 1 h. Rabbit primary antibodies raised against 4mC- or 6mA-modified DNA[77] were a kind gift from Dr. Iain Murray (NEB), and were re-checked for the absence of cross-reactivity, as well as for lack of reactivity with 5mC on human DNA. The membrane was washed three times with TBST, incubated with 1:10,000 goat anti-rabbit horseradish peroxidase (HRP) antibody (Sigma A0545) at room temperature for 1 h, washed three times with 1× TBST, and developed using SuperSignal™ West Dura Extended Duration Substrate (Thermo Fisher Scientific). Chemiluminescence was detected using the Amersham Imager 600 (GE Healthcare).

**Electrophoretic mobility shift assays**. sAvL1-451 DNA were 5′-end-labeled with [γ−32P]dATP (PerkinElmer) using T4 polynucleotide kinase (NEB) and purified from excess of radioactive nucleotides using Oligo Clean & Concentrator kit (Zymo Research) following the manufacturer's protocols. Binding reactions were set up in 10 μl total volume in a buffer with final concentrations 100 mM KCl, 10 mM Tris, pH7.4, 0.1 mM EDTA, 0.1 mM DTT, supplied with 500 ng LightShift™ Poly (dI-dC) (Thermo Scientific). Addition of 2.5 μl of AvMBD proteins provided 5% glycerol per reaction. Proteins were first pre-incubated with non-radioactive DNA for 15 min at RT. Then, 32P-labeled DNA was added to a final concentration of 0.05 nM, and reactions were incubated for additional 30 min at RT. After supplying with 6× EMSA gel-loading solution (Thermo Scientific), samples were loaded onto 6% DNA Retardation gels. Samples were run at 90 V in 0.5× TBE buffer (44.5 mM Tris–HCl, pH 8.3, 44.5 mM boric acid and 1 mM EDTA) at 4 °C for 90 min. Gels

were dried using Model 583 Gel Dryer (BioRad), exposed with phosphorimaging plate (Fujifilm), scanned on Typhoon FLA 7000, and analyzed using Image Quant TL v8.1 software.

**DNA extraction for DIP-seq**. For genomic DNA extraction, animals were starved for 48 h and treated with ampicillin and tetracycline antibiotics at final concentrations of 10 mg/ml and 0.5 mg/ml, respectively, for 24 h, then harvested as described in ref. [17]. Total DNA was extracted with DNeasy Tissue kit (Qiagen); eluates were checked by agarose gel electrophoresis and final concentrations were measured by Nanodrop. The isolated genomic DNA was diluted to ~250 ng/μl using TE buffer and sonicated on the 130 μl scale (Covaris microtubes) to 200–400 bp using Covaris S220 focused ultrasonicator (10% duty cycle, 175 W peak, 200 cycles, 180 s, 6 °C). After measuring concentration and size distribution with Bioanalyzer High Sensitivity DNA chip (Agilent), 100 ng of fragmented DNA was used for library construction with NuGen Ovation Ultra-Low System v2.

**DIP-seq (MeDIP-seq)**. After adaptor ligation and purification steps (NuGen Ovation Ultra-Low System v2 protocol), DNA fragments were combined with 0.5 μg of anti-4mC or anti-6mA antibodies (see above) in 500 μl of 1× IP buffer and incubated at 4 °C for 6 h. In parallel, 40 μl of Protein A magnetic beads were prepared as in ref. [78]. Protein A beads were added to DNA–antibody mixture and incubated at 4 °C overnight with rotation. Beads were washed four times with 1× IP buffer on a magnetic rack. Proteinase K (20 μl of 20 mg/ml solution) was used to release the methylated DNA with 3 h of incubation at 50 °C. The final eluate was purified using 2× phenol–chloroform–isoamyl alcohol (25:24:1) extraction and ethanol precipitation. DNA was resuspended in 35 μl H$_2$O, followed by library amplification and bead purification (NuGen RNAClean XP magnetic beads). Quality control and concentration measurement were performed using Bioanalyzer DNA 1000 chip (Agilent) and Qubit sDNA HS Assay kit (Thermo). Libraries were sequenced using the Illumina HiSeq 2500 platform (50-bp SR) at the Brown University Sequencing Core Facility. Base-calling was performed with the standard Illumina pipeline (Casava 1.8.2). Illumina adaptors were trimmed with cutadapt v1.9.2[79], as well as any sequence with low quality score (<Q20) and/or <16 nucleotides in length (FASTX v0.0.13 Toolkit). Reads were aligned to Av-ref[17] and AvL1 assembly (see below) using Bowtie v1.1.0[80], with parameters permitting less than one mismatch in the first 30 bases. MACS v1.3[81] was used to locate enriched regions for 4mC and 6mA in both genomes, using nomodel nolambda parameters.

**Genome assembly**. The initial *A. vaga* L1 draft assembly was generated with high quality paired-end Illumina MiSeq reads using SPAdes assembler to yield N50 of 18.125 kb[35]. However, the published AvL1 assembly filtered any sequences without blastn matches to Av-ref, which may include recent horizontal transfers and strain-specific TEs. To improve the initial assembly, DNA was extracted from rotifer eggs as in[42], and a 20-kb library was constructed using BluePippin selection to sequence 15 SMRT cells on a PacBio RS II sequencer (Pacific Biosciences) at the Johns Hopkins University Deep Sequencing and Microarray Core facility with P6-C4 chemistry (accession number PRJNA558051). We used PBJelly from PBSuite 15.8.24[82] with PacBio filtered subreads to improve the initial AvL1 assembly. A total of 890,504 PacBio subreads with N50 read length of 16,294 bp was used after SFilter (Pacific Biosciences) and spike-in control removal. The improved hybrid assembly was filtered from contaminants using bacterial single-copy genes, GC-content, k-mer frequencies ($k = 4$), and DNA coverage values (both Illumina and PacBio), as in[83]. Assembled contaminant contigs, mostly of bacterial origin, were filtered out to yield a final assembly totaling 217.1 Mb in 9856 contigs (Supplementary Table 5), which is very close to the 218-Mb Av-ref assembly[17] and improves by 20 Mb the Illumina-only assembly, increasing N50 from 22.1 to 87.4 kb. We also identified 12 chimeric contigs, listed in Supplementary Data 4, which were mostly eukaryotic with an attached small stretch of bacterial DNA displaying high methylation density. The AvL1 assembly used in this work has been deposited at DDBJ/ENA/GenBank under the accession JAGENE000000000. The version described in this paper is version JAGENE010000000. Its accession number is GCA_021403095.1 in the NCBI Assembly database. Although a chromosome-scale *A. vaga* assembly is expected to become available soon[84], the current bdelloid assemblies display adequate levels of contiguity to examine nearly all genomic regions.

**PacBio modification analysis**. We examined genome-wide distribution of modified bases in SMRT-seq data[85] with SMRT Analysis Software 2.3.0. Raw data from 15 AvL1 SMRT cells were filtered by SFilter (Pacific Biosciences) to remove reads containing adapters, short reads and low-quality reads with cutoffs for read quality ≤ 0.75, read length ≤ 50 nt, and subread length ≤ 50 nt. Filtered reads were aligned to the AvL1 assembly using RS_Modification_Detection.1 protocol (Pacific Biosciences). Briefly, the cleaned reads were aligned to AvL1 curated genome assembly using blasr v1.3.1[86]. The polymerase kinetics information was processed and reported as IPD ratio, with its fraction (the methylated portion of reads mapped) at each site. The 4mC and 6 mA base modifications were identified, and the final report was extracted as csv and gff files for posterior processing. Filtering was performed by selecting only 4mC and 6mA marks with 20× coverage and mQv ≥ 22 (Supplementary Table 6); any sites with coverage <10× were removed.

Although SMRT analysis may sometimes erroneously identify 5mC as 4mC, as occurred for the fig genome[87], which has a full complement of plant 5mC-MTases but no N4C-MTases, we are confident that multiple orthologous methods applied to *A.vaga*, which lacks 5mC-MTases but has the N4C-MTase, validate our SMRT-seq cytosine modification calls as 4mC. Methylation fraction values were converted into bigwig file format and plotted with deepTools2[88]. Methylation fractions for DIP-seq peak summits and transposons were represented per annotation, with the y-axis as "Mean normalized fraction". Additional analyses were done with custom scripts for plotting results with R. We separated 4mC and 6mA according to their methylation levels: low-fraction sites (0.1–0.5), moderately methylated (0.5–0.8), and highly methylated (0.8–1). The upstream and downstream 10-bp sequences from 4mC and 6mA modification sites were extracted for motif identification in each group by MEME-ChIP v5.4.1[89]. The nucleotide adjacent to the methylated sites was pulled out for counting the proportion of doublets.

**Dot-blot immunoassays for histone marks**. We first assayed, by dot-blot analysis, the reactivity of *A. vaga* histone methylation marks with Premium ChIP-seq grade affinity-purified rabbit polyclonal antibodies H3K4me3, H3K9me3 and H3K27me3, raised against synthetic peptides with the corresponding trimethylated lysines (Diagenode C15410003, C15410056 and C15410195, respectively). These antibodies display reactivity with a wide range of species including vertebrates, *Drosophila*, *C. elegans* and plants, and have been tested by ChIP-seq, IF, Western blotting, and ELISA. The H3 N-terminal residues 1–31 display 100% identity between *A. vaga* and humans; although formally cross-reactivity of K9/27 cannot be excluded for *A. vaga*, none was observed in human peptide arrays spanning the identical aa sequence (Diagenode). Protein extracts from Av-ref and AvL1, resuspended in 0.5 v of extraction buffer (10 mM Hepes, 5 mM MgCl$_2$, 2 mM DTT, 10% glycerol and cOmplete protease inhibitor tablets (Roche)), were spotted on BioTrace™ NT Nitrocellulose Transfer Membrane (Pall Corporation), air dried and blocked in 5% BSA in TBST (containing 0.05% v/v Tween) for 1 h at RT and incubated with 1:10,000 anti-H3K4me3, H3K9me3 or H3K27me3 antibodies at RT for 1 h. The membrane was washed three times with TBST, incubated with 1:10,000 goat anti-rabbit HRP antibody (Sigma A0545) at room temperature for 1 h, washed three times with 1x TBST, then once with TBS and developed using SuperSignal™ West Dura Extended Duration Substrate (Thermo Fisher Scientific). Chemiluminescence was detected using the Amersham Imager 600 chemiluminescence imager (GE Healthcare).

**ChIP-seq**. Chromatin immunoprecipitation (ChIP) was performed based on the *C. elegans* protocol[90] with minor modifications. Briefly, rotifers were starved for 48 h before collection, and live animal pellets were washed with PBS, followed by another round with protease inhibitor (cOmplete Roche tablet). The 1-ml pipette tip was used to drip the mix into a porcelain mortar containing liquid nitrogen, and the frozen rotifer "popcorn" was ground to fine powder with a pestle. Nuclear proteins were cross-linked to DNA by adding 1.1% formaldehyde (Thermo) in PBS + 1x protease/phosphatase inhibitors (Halt™ Protease & Phosphatase Inhibitor Cocktail, Thermo) for 10 min at room temperature on a rocking platform. Cross-linking was stopped by adding glycine to a final concentration of 0.125 M and incubating for 5 min at RT. The medium was removed, and the cells were washed twice with ice-cold PBS containing 1 mM PMSF. The cells were then collected in FA lysis buffer (FA buffer + 0.1% sarkosyl + protease/phosphatase inhibitors); FA buffer: 50 mM HEPES/KOH pH 7.5, 1 mM EDTA, 1% Triton™ X-100, 0.1% sodium deoxycholate, 150 mM NaCl. Subsequently, the chromatin was isolated, sonicated (Covaris S220: 2% Duty Cycle, 105 W Peak, 200 Cycles, 360 s, 6 °C), and immunoprecipitated with 1 μg anti-H3K4me3 antibody, anti-H3K27me3 antibody, or anti-H3K9me3 antibody (all from Diagenode as above) or no antibody (input control). After reverse-cross link (overnight at 65 °C), DNA was purified by using 2x phenol-chloroform-isoamyl alcohol (25:24:1) extraction and ethanol precipitation. DNA was resuspended in 35 μl 10 mM Tris-Cl, pH 8.5. The ChIP DNA and input DNA were used to construct ChIP-seq libraries using NEBNext Ultra II DNA Library Prep Kit (NEB) following the manufacturer's procedure. Libraries were sequenced on Illumina NextSeq 500 platform for 75 bp single-end HT at the W.M. Keck Sequencing Facility at the MBL. After demultiplexing and adapter trimming (bcl2fastq software, Illumina), raw reads were cleaned up to obtain high-quality reads (see parameters in IP-seq). Clean reads were mapped to Av-ref and AvL1 assemblies using bowtie2 v2.2.5[91] with default parameters. Genomic regions associated with histone modification were identified using Model-based Analysis of ChIP-Seq (MACS2 v2.1.0)[81] using default parameters.

**RNA-seq**. For *A. vaga* Av-ref transcriptome, total RNA was extracted from animals at all life-stages with TRIzol® (Invitrogen) following manufacturer's protocol with a glass Dounce homogenizer. After DNase I (NEB) treatment on RNA Clean & Concentrator columns C-5 (Zymo Research), *A. vaga* total RNA was eluted and subjected to poly-A selection with Ambion MicroPoly(A) Purist Kit (Thermo Fischer). Libraries were prepared with Encore Complete Library RNA-Seq Library Systems (NuGen). A total of 3 biological replicas were sequenced on a dedicated Illumina NextSeq Mid lane (1 × 150 bp) and, after QC (http://hannonlab.cshl.edu/fastx_toolkit/) and adapter trimming (Cutadapt v1.9.2)[79], mapped to Av-ref[17] with Tophat 2.1.1[92], using default parameters

and –max-intron-length 100. Aligned reads were counted by genomic feature with HTSeq-count v0.6.1[93], using default parameters.

For AvL1 transcriptome, RNA extraction was performed following[17] for the fully hydrated *A. vaga* L1 cultures containing animals at all life stages. Rotifers were collected by centrifugation at 4000*g*. After removal of the supernatant (spring water), total RNA was extracted with Trizol (Invitrogen) followed by ethanol precipitation. After DNaseI treatment (DNA-free, Ambion), 1 μg of total RNA was shipped for QC, library preparation (eukaryotic mRNA protocol), and Illumina sequencing (HiSeq x PE150 bp) to Novogene Co., Ltd. Raw reads (~3.3 Gb) from two lanes as technical replicates were processed (see parameters in IP-seq), and properly paired reads were aligned to the AvL1 assembly using TopHat v2.1.1[94], using default parameters and –max-intron-length 100. Mapped reads were counted within each feature with HTSeq-count[93] using default parameters, which were used to calculate RPKMs of annotated genes.

**Prediction of protein-coding genes**. BRAKER v2.1.2[95], a combination of GeneMark-ET[96] and AUGUSTUS[97], was used to predict protein-coding genes in the AvL1 genome using aligned RNA-seq data. TopHat alignments were used to generate UTR training examples for AUGUSTUS to train UTR parameters and predict genes. This procedure was done with –soft masking enabled, after masking the genome with RepeatMasker v4.0.7 (see *Repeat annotation*). Total predictions comprised 74,569 gene models originating from 74,233 loci. Initial predictions were filtered from TE genes using AvL1 TE annotations (RepeatMasker) and BLAST homology search to known TE proteins. BLAST searches were performed with 74,569 gene predictions using blastp (blast+) and blastx (diamond blast) onto nr and uniref90 databases, respectively. BLAST descriptions with TE-related terms ("transposon", "transposable", "integrase", "reverse transcriptase", "pol", "gag") were considered as TE-associated proteins. A total of 977 genes were classified as AvL1 TE-related. A further quality check of gene annotations filtered incomplete genes. Annotations at the contig boundaries were removed ($n = 5205$), along with CDS that carried a premature stop codon ($n = 282$) or without appropriate termination codon at the CDS end ($n = 2748$, which mostly fall on contig boundaries). A final filter was applied to remove annotations with no BLAST homology (neither nr nor uniprot) and for which CDS sequence was under 300 bp. A final gene set of 65,934 annotations was used for downstream analysis.

**Repeat annotation**. We used the REPET package with default settings for initial AvL1 de novo TE identification and annotation[98]. The automated library of TE families was subjected to extensive manual curation, as was previously done for Av-ref[17], and used as a database for searching and annotating TE copies in the AvL1 assembly with RepeatMasker[99]. We used RMBlast (National Center for Biotechnology Information Blast modified for use with RepeatMasker) as a search engine. Initial RepeatMasker output was filtered for copies covering less than 5% of reference TE length and converted into gff3 format for subsequent analysis. TE annotation was intersected with gene models to eliminate any duplication events spanning both databases and to obtain a list of TE-encoded genes for further analysis. For TR identification, AvL1 assembly was uploaded to TRs Database[100]. We generated an initial set of TRs by analyzing the sequence of each contig using Tandem Repeats Finder v4.09[101] with default parameters (match = 2, mismatch = 7, indels = 7, minimal alignment score = 50). Further searches with modifications in the alignment score (size of the repeat unit) were performed, and manual correction was carried out when necessary.

**Small RNA analysis**. *A. vaga* sRNA-seq data (SRA accession no. SRP070765) for two wild-type small RNA replicas were mapped to Av-ref genome as described in ref. [50]. Heatmaps of sRNA-seq data for genes, TEs, and DIP-seq and ChIP-seq peaks were generated with deepTools2[88] for each annotation. Reads normalized to 1× sequencing depth (RPGC, reads per genomic content) were used for normalization in heatmaps.

**Methylation data processing and visualization**. For generation of heat maps and profile plots, the deepTools2[88] computeMatrix, bamCoverage, bamCompare, plotHeatmap and plotProfile scripts were used with specific parameters: RPGC normalization, bin size 10, effective genome size (Av 213837663 and AvL1 217117546), extendReads (IP-seq 50, ChIP-seq 75, sRNA-seq 50), interpolationMethod nearest. Methylation profiles for DIP-seq/ChIP-seq were represented per annotation as mean normalized tag signal, with the *y*-axis labeled "IP/ChIP occupancy". While using input DNA for comparison, the profile is represented as mean normalized log2 ratio, with the *y*-axis labeled "log2 ratio". Methylation profiles for DIP-seq were represented per annotation, with the *y*-axis labeled "IP occupancy". The annotatePeaks function from HOMER Tools v4.11[102] was used to obtain methylation profiles of selected regions of interest, using different window and bin sizes (parameters given in figure legends). Overlapping values of different annotated features (DIP/ChIP-seq peak, base modification) were estimated with bedtools v2.27.1[103], whether they are intersecting (bedtools intersect) or after providing a specific size window (bedtools window). Genome-wide 4mC/6mA visual representations were generated using Circos v0.69-6[104] as follows: Av-ref reads were plotted from two genomic Illumina libraries (SRP020364) with different insert size (450 and 862 bp); AvL1 reads were plotted from Illumina (SRR8134454) and

PacBio (SRX6639068); RNA-seq reads were plotted from SRP228822 (Av-ref and AvL1); and Av-ref small RNA reads from SRP070765. Additional visual representations for selected contigs were obtained with IGV viewer[105].

**Collinearity analysis**. Syntenic regions within and between genomes were identified using MCScanX v0.8[106] after blastp all-versus-all (e-val = 1e−10, maximum number of target sequences = 5) of the protein annotations from both genomes (Av-ref and AvL1). We searched for collinear block regions with at least 3 homologous genes and 20 maximum gaps allowed. The Ks and Ka (synonymous and nonsynonymous substitution, respectively) values between pairs of collinear genes were calculated with the script add_kaks_to_MCScanX.pl (https://zenodo.org/badge/latestdoi/92963110). We also searched for collinearity breaks between adjacent homologous blocks, defined as regions where homologous blocks could not be aligned along scaffolds without some rearrangements.

**Phylogenetic analyses**. MTase homologs in bdelloids were identified by tblastn searches of GenBank WGS databases at NCBI, checked for the presence of metazoan genes in the vicinity, translated with validation of exon–intron structure, and used in blastp searches of REBASE (http://rebase.neb.com/rebase/)[1] to obtain MTases with known recognition sequences. Multiple sequence alignments were performed with MUSCLE v.3.8.31[107] and manually adjusted when necessary. Amino acid sequences were clustered by neighbor-joining, as MTases are not amenable to conventional phylogenetic analysis due to hypervariability of the target recognition domain, and the tree was visualized in MEGA[108]. MBD-containing bdelloid proteins were identified by profile HMM search[109] with the MBD query (PF01429). Av-ref SETDB1 homologs from the Genoscope annotation were manually re-annotated to improve quality, and full-length proteins were used as queries in blastp searches of refseq_protein database at NCBI to obtain additional orthologs from 10 bdelloid species and representative protostome taxa. Maximum likelihood phylogenetic analysis was done with IQTREE v1.6.11[110] using best-fitting model selection and 1000 ultrafast bootstrap replicates. Ago/Piwi counts in AvL1 were done as in ref. [51].

**Reporting summary**. Further information on research design is available in the Nature Research Reporting Summary linked to this article.

## Data availability

The data that support this study are available from the corresponding author upon reasonable request. Sequences obtained in this study were deposited under BioProject PRJNA558051 (SRA accession Nos. SRR9886612, SRR9900832-45 for individual SMRT cells). Avaga_MBL_L1 genome assembly project was deposited in DDBJ/ENA/GenBank under the accession JAGENE000000000. The version described in this paper is version JAGENE010000000. Its accession number is GCA_021403095.1 in the NCBI Assembly database. ChIP-seq, MeDIP-seq and RNA-seq data generated in this study have been deposited in the GEO database under accession code GSE140049, GSE140050, GSE140051, and GSE140052. All materials are freely available without restrictions. Source data are provided with this paper.

## Code availability

Custom scripts are available through GitHub.

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

## Acknowledgements

We thank Iain Murray (NEB) for the kind gift of anti-4mC and anti-6mA antibodies and Michael Shribak (MBL) for polarization microscope imaging. This work was supported by grant R01GM111917 from the U.S. National Institutes of Health to I.A.

## Author contributions

F.R. was responsible for high-throughput genomic and transcriptomic data generation and analysis. I.Y. performed protein expression, purification, biochemical characterization, and analysis. D.D. conducted pilot experiments at the early stages of this work. I.A. conceived and designed the project, analyzed the data, and drafted the paper. All authors contributed to writing and editing the final version.

## Competing interests

The authors declare no competing interests.
