## [Peer Review File · Nature Communications]

REVIEWER COMMENTS

Reviewer #1 (Remarks to the Author):

The Editor invited me to comment on the data analysis part of the manuscript. The Editor may know that my expertise on algorithm design and machine learning theories are far away from the main topics/supplementary topics of the research.

I looked the subsections in the Methods section. What I can understand is only about the subsections: Genome assembly, Prediction of protein-coding genes and phylogenetic analysis.

From my understanding, the author has taken existing tools for genome assembly, prediction of protein-coding genes and phylogenetic analysis. For example, the genome assembly approach is the commonly used hybrid approach, combining de novo assemblies from Illumina short reads with those from PacBio long reads-based assemblies for joint polishing of the final genome. This approach is very advanced in the field.

Prediction of protein-coding genes is also by an existing tool named BRAKER, a combination of GeneMark-ET and AUGUSTUS. The steps are used correctly, for example the construction of training sets.

Except from this handful of technical comments, I feel that the paper is well organized with clear contribution presented at each section.

Reviewer #2 (Remarks to the Author):

This manuscript reports the important discovery of the acquisition of a bacterial method of DNA modification by a eukaryote via horizontal gene transfer. This discovery is interesting because it sheds light on the process of gains in DNA modification mechanisms, which otherwise happened much deeper in the evolutionary past. The authors have taken great pains to investigate the mechanisms and consequences of this change with detailed molecular investigation. This makes the manuscript quite complex and 'full', but the more general sections do a good job of explaining the context and importance.

One part that is hard to follow for a non-expert and could be improved is the abstract. An initial sentence setting the scene would be useful, i.e. that you are talking about epigenetic modifications to DNA that influence gene expression. Then the statement of the general pattern in eukaryotes and bacteria is in terms of the modified base, but the statement of the enzymes they lack and possess is not easily connected to this earlier sentence for a non-expert, i.e. that C5-methyltransferase is the enzyme that modifies 5-methylcytosine. Line 22 has syntax "histone-read-DNA-write" that is abbreviated and not easily interpretable, and line 23 has a very complex enzyme name and compact description of effects on transposons. I appreciate this is a difficult topic, but I think the abstract could be improved to pull out the general points more clearly. The introduction is much clearer and perhaps a summary of some of this material would help the abstract: "Dna modification is used to regulate expression and in defence against transposable elements. In eukaryotes it normally involves X, in bacteria Y. We find that bdelloid rotifers lack X and have acquired mechanism Y by HGT from bacteria, and deploy this in defence against transposable elements".

Lines 198 to 211 is rather complex to say that 4mC show same pattern as in Av-ref, but that 6mA show a different pattern and are also positively associated with TEs in AvL1. Maybe the detail could move to supplementary and just say what is the same and different.

Line 230. Why did you use eggs for this part?

Line 350. I lost track of what this entire section is for. Why is it interesting whether they show in vitro activity? What is the take-home message for this whole section?

Curious that both bdelloids and monogononts have lost the C5-MTases, as well as nematodes and tardigrades. What is the hypothesis for why they lost that?

Reviewer #3 (Remarks to the Author):

In this work, Rodriguez and colleagues aim to demonstrate that bdelloid rotifers present N4-methylcytosine in the genome (4mC). To test this, they identify a putative 4mC methyltransferase that has no close homology to eukaryotic sequences. Then perform DIP-seq (antibody-based immunoprecipitation followed by sequencing) and use PacBio reads to identify 4mC and 6mA in the genomes of two very closely related "species" of *Adineta vaga* rotifers. Then express the putative methyltransferases in vitro and in bacteria to verify the methyltransferase activity. Additionally, the authors perform ChIP-seq of three histone-tail modifications (H3K4me3/H3K9me3/H3K27me3) to understand the possible crosstalk with 4mC (and 6mA). Finally, they try to identify a putative reader of 4mC in the *Adineta* genome, and perform in vitro experiments to test the capacity of a Methyl Binding Domain protein to perform this function.

Overall, it would be an exciting finding to show that 4mC is present in rotifer genomes. This manuscript presents a compelling case for this being the case, but offers quite weak support for the genomic distribution of 4mC in the genomes of these species or its possible functions. Then, many of the other claims raised across the manuscript are also weak. The manuscript is unfocused and too long, difficult to follow in many sections, and should be shortened drastically, with some sections dropped. Also, the methods used to analyse a lot of the data in this manuscript are not aligned with the standards of the field. In sum, I would not recommend the publication of this manuscript in the current form in *Nature Communications*.

The first criticism in terms of novelty is that *Marchantia polymorpha*, a land plant, has been reported to contain 4mC recently (<https://www.biorxiv.org/content/10.1101/2021.02.12.428880v1>). In the *Marchantia* work, the authors identify the responsible 4mC methyl-transferases and are able to profile 4mC by using whole genome bisulfite sequencing and SMRT-seq, to achieve precise base-resolution identification of 4mC sites (unlike in this manuscript). That work does not imply the findings in rotifers are not meaningful and potentially very novel, but the authors should discuss that work throughout the manuscript and avoid claims such as this being the first example of 4mC in a eukaryotic genome.

In general, I believe that the methyltransferases identified in this manuscript are possibly active in rotifers, and they are likely originated from bacterial LGT (although the sister group in the tree are cyanophages, which supports a viral origin). But the distribution of 4mC (and 6mA) is way less clear, making the function of these modifications dubious in rotifers. DIP-seq is well known to be prone to artifacts, and the lack of coherence between the DIP-seq datasets between the two genomes used in this manuscript clearly shows that. Also, the analysis of "peak coverage" used in the figures is very unusual, this type of data, like ChIP-seq, is shown as relative enrichment versus input (IgG control), or directly as normalised read coverage. Then, all the SMRT-seq data is also shown as "coverage", when methylation data across the literature is always shown as a fractional value. Methylation goes from 0 to 1 on a given base depending on the proportion of reads that support methylation, and that is what is informative when showing "metaprofiles" of any methylation mark. This is not done in this manuscript, and renders it very difficult to interpret. Also, the methods used by the authors do not

show coherence between DIP-seq and SMRT-seq, which makes the whole claim that these marks are enriched on transposable elements dubious.

Including the 6mA data is also quite problematic. The presence of 6mA is highly contested in animal genomes, and the data presented here is not robust. 6mA could be RNA contaminations (as found in mammalian species). The claims regarding this section do not offer much novelty and remain contentious.

Would be good to show the intron-exon structure of the gene with RNA-seq data, and presenting some extra quality checks ensuring these genes are not found in bacterial scaffolds and are indeed encoded as host genes.

Since both genomes are so close, one way to make the claims more robust would be to show orthologous regions across both species with similar profiles in terms of SMRT-seq base modification and DIP-seq.

Figure 1f contains some errors. There is no need to establish "Phylum (Class)". This is not very useful, plus, the authors do not stick to this classification for many groups. E.g. Fungi are not a Phylum. "Protist" is not a valid category, Amoebozoans are more closely related to Animals and Fungi than to Ciliates. Similarly, Ciliates should be sister group to the plant lineages. "Higher plants" does not have any meaning, there is no "lower" or "higher" plants, please refer to the correct group (Embryophytes?). Early Metazoa -> Non bilaterians. Green algae -> Chlorophyta? Some of the common names are confusing.

Figure 2 is a mess. Why there are 2 letter codes? Some of them are in capital, others in lowercase, very confusing.

Figure 2a. What does "coverage" represent? Peaks? What is "peak" coverage? Number of peaks that intersect with that window? That is not how these metaprofile plots are usually computed. These take the aligned reads from the DIP-seq (or ChIP-seq) and transform it to a read coverage plot, corrected by coverage (e.g. Counts per Million), and, if available, input DNA (IgG only). Those are the values then used to compute "coverage" plots such as these. The authors have used DeepTools in Extended Data Fig 2 c/d, but that figure is rather unclear. First, it shows enrichment in all features for both 4mC/5mC. It also says "Relative Fold Enrichment" (respect to what?). TS should be TSS (Transcriptional Start Site), and TTS should be TTS (Transcriptional Termination Site), these are standards in the field, why change them? DeepTools plots can be modified in Inkscape or Illustrator, that would be highly beneficial, since "Gene distance" is not very intuitive when representing Transposable Elements.

The Figure 2a "coverage" metaprofiles do not fit at all with those shown in Extended Data Fig 2a/b. This highlights how unreliable are these DIP-seq techniques, which show very different results in two very closely related rotifers.

Figure 2d. I believe the SMRT analysis software provides motifs enriched on methylated bases, why did the authors show MEME-ChIP which clearly does not fit this data?

Figure 2g and h. PacBio counts should not be used, methylation levels are always shown as fractional levels since this is a technique with base resolution.

Figure 2i indicates that Adineta has enriched 4mC and 6mA on transposable elements, when other plots in this same figure indicate that 6mA is not enriched on these elements. This is very confusing. Also, showing "dynamite plots" is bad practice, these values should be shown as a distribution (boxplot?). I still do not know why SMRT counts are used instead of fractional values.

Figure 2c. What is "coverage"? PacBio base modification relies on coverage on positions, but then gives fractional values on any given position (0 to 1 values). In fact, "coverage" is a meaningless feature to determine the level of a base modification in any given position, but the "fraction methylation levels" should be averaged and displayed for a metaprofile like this. Again, I would recommend the authors to clarify what this is and maybe use DeepTools2. Why does it say 2.5 Kb window? The plot shows 3kb. Same for the one below, 500 bp window but there are 600 bp shown in the x axis.

Extended Data Fig 3 is very hard to read, pixelated when zoomed in. There is not need to show these are "Circos" plots, simple Genome Browser representations would be much easier to interpret, zooming out or in depending on the features that need to be highlighted.

The sequences of the conserved motifs shown in Figure 3f do not contain any CG dinucleotide, which in figure 2b was shown to be the preferred substrate of these enzymes according to SMRT-seq data. This is inconsistent and is not discussed.

How are the + and - in Figure 3g determined? With an immuno-dot blot? Where is it? Not all combinations seem to be found in h and i panels. H and I panels are hard to interpret and poorly explained in the figure legend.

Figure 4. H3K9me3 and H3K27me3 are not found usually on the same sites, one is constitutive chromatin and the other facultative (at least in mammals). This "expectation" is not very realistic. To validate the ChIP-seq, it would be good that the plots in Figure 4a/c would be sorted the same way, e.g. using the one to one orthology relationships between assemblies/species. Also, it would be good to add more information on the figure itself, reading the legend is confusing.

Figure 4e/f. This plot is suspicious, the shape of the "peak" indicates that there is unclear smoothing within that window. Usually peak metaprofiles are centred around the peak summit (or centre), without setting hard borders as if they were genes, which tends to generate artefactual shapes as those shown here. Also, figure legends are barely visible.

Figure 4h. Almost impossible to read, too pixelated. Better to show linear genome browser snapshots highlighting some associations instead of dumping too much data in a circular plot that is difficult to follow.

Figure 4g. This should resemble figure 4f using a metaprofile displaying the fractional level of methylation (4mC/6mA) on those peaks, and not a "dynamite plot" that is difficult to interpret.

The section on the small RNAs feels confusing and unnecessary for this manuscript.

The section about SETDB1 as a possible reader of 4mC is also very weak and unnecessary. Not all animal MBD proteins bind to methylated cytosines, that is well established. MBD1/2/3/4 and MeCP2 are the families that have been shown to recognise methylated CpGs in other animals. In some cases, like *Drosophila melanogaster*, it harbours a highly divergent MBD2/3, that has been shown not to bind to preferentially to methylated CpGs, which is expected since *Drosophila* lacks 5mC methylation. The competition experiments shown in Figure 6 are not very strong indication of this domain preferentially binding to 4mC methylation.

Figure 6f. Given the data shown in this manuscript, it feels premature to draw a model on how this system works in tardigrades.

Claims such as "Finally, it demonstrates that horizontally transferred genes, contrary to the established view". Lateral Gene Transfer is contested as a major source of innovation in eukaryotic lineages by some authors, but this is far from being "the established view".

Plots such as those in Extended Data Fig 5 e,f,g,i are basically noise. This is likely due to the non-standard ways of representing this data, and no information can be extracted from these.

Reviewer #4 (Remarks to the Author):

Comments to the author

Rodriguez et al. performed an impressive research on 4mC occurrence in eukaryotic DNA. The major result underlines the 4mC presence in bdelloid rotifers by combining multiple trials I believe the results reported would be interesting to wide spectra of biologist and other specialists.

Points raised:

1. In figure 1a,1b, is it more intuitive to mark the direction with the N-terminal or C-terminal instead of using numbers to mark the direction? In addition, using cylinders to roughly represent multiple enzymes seems insufficient. Is there a way to show the structural differences more clearly? I'm not whetted on it, but think the article would be improved, if the authors consider this and take some actions.
2. The author mentioned that for gene profiles, the modification density is much lower, and it seems the opposite compared to TE profile. Can the authors explain the opposite distribution phenomenon of gene profile and TE profile in a more understandable way?
3. As far as I know, coverage level is a major effect factor on correction efficiency. In the range of 10x to 50x, the correction result increases as the coverage increases (https://www.uni-wuerzburg.de/fileadmin/07030400/AG_Genomics/Proovread/proovread-preprint.pdf). Therefore, does the author think that after increasing the coverage level, the overlap between SMRT-seq and DIP-seq is higher?
4. In line 247, the author mentioned "At 4mC sites, CpG and CpA dinucleotides are the most prevalent, making up 74% of modified doublets." Please explain the calculation process.
5. In line 259, the author reports the different methylation levels of 4mC and 6mA. Whether the conclusion drawn is related to the state of the sample. In other words, the methylation levels show differences at different time nodes.
6. Please use the definite article accurately and polish other details that I haven't noticed. For example, "the genome-wide", "We visualized the distribution", etc.
7. The ordinate of Fig. 2B is not clear, and the picture resolution needs to be improved.
8. It may be due to the typesetting that makes Figure 2 look confusing. Please adjust the position and label of the picture to make it easier for readers to read. In addition, m4C, m6A in the 2F should be modified to 4mC, 6mA, please keep the abbreviations consistent.
9. Some typos should be revised.

Reviewer #1 (Remarks to the Author):

The Editor invited me to comment on the data analysis part of the manuscript. The Editor may know that my expertise on algorithm design and machine learning theories are far away from the main topics/supplementary topics of the research.

I looked the subsections in the Methods section. What I can understand is only about the subsections: Genome assembly, Prediction of protein-coding genes and phylogenetic analysis.

From my understanding, the author has taken existing tools for genome assembly, prediction of protein-coding genes and phylogenetic analysis. For example, the genome assembly approach is the commonly used hybrid approach, combining de novo assemblies from Illumina short reads with those from PacBio long reads-based assemblies for joint polishing of the final genome. This approach is very advanced in the field.

Prediction of protein-coding genes is also by an existing tool named BRAKER, a combination of GeneMark-ET and AUGUSTUS. The steps are used correctly, for example the construction of training sets.

Except from this handful of technical comments, I feel that the paper is well organized with clear contribution presented at each section.

We thank the reviewer for the positive assessment of our genome assembly and annotation approaches and of the manuscript in general.

Reviewer #2 (Remarks to the Author):

This manuscript reports the important discovery of the acquisition of a bacterial method of DNA modification by a eukaryote via horizontal gene transfer. This discovery is interesting because it sheds light on the process of gains in DNA modification mechanisms, which otherwise happened much deeper in the evolutionary past. The authors have taken great pains to investigate the mechanisms and consequences of this change with detailed molecular investigation. This makes the manuscript quite complex and ‘full’, but the more general sections do a good job of explaining the context and importance.

One part that is hard to follow for a non-expert and could be improved is the abstract. An initial sentence setting the scene would be useful, i.e. that you are talking about epigenetic modifications to DNA that influence gene expression. Then the statement of the general pattern in eukaryotes and bacteria is in terms of the modified base, but the statement of the enzymes they lack and possess is not easily connected to this earlier sentence for a non-expert, i.e. that C5-methyltransferase is the enzyme that modifies 5-methylcytosine. Line 22 has syntax “histone-read-DNA-write” that is abbreviated and not easily interpretable, and line 23 has a very complex enzyme name and compact description of effects on transposons. I appreciate this is a difficult topic, but I think the abstract could be improved to pull out the general points more clearly. The introduction is much clearer and perhaps a summary of some of this material would help the abstract: “Dna modification is used to regulate expression and in defence against transposable elements. In eukaryotes it normally involves X, in bacteria Y. We find that bdelloid rotifers lack X and have acquired mechanism Y by HGT from bacteria, and deploy this in defence against transposable elements”.

Thank you for the valuable comments. We have reworded the beginning of the abstract using the suggested summary, to make it clearer for non-experts. We removed the extra “eggless” component from the enzyme name, limiting it to the more widely recognized SETDB1. The 150-word limit prevents us from fully deciphering the “read-write” syntax, which we believe is simplistic enough and is often used to facilitate understanding (it is used even in titles, e.g. ref. #70 Park et al. 2019).

Lines 198 to 211 is rather complex to say that 4mC show same pattern as in Av-ref, but that 6mA show a different pattern and are also positively associated with TEs in AvL1. Maybe the detail could move to supplementary and just say what is the same and different.

We have moved the details to the Supplementary Note 1, which helped to shorten the main text.

Line 230. Why did you use eggs for this part?

The advantage of using eggs to generate PacBio data was twofold: to discern methylation patterns that are most stable throughout all stages of development, and to increase PacBio coverage by minimizing bacterial contamination, which was achieved by washing and Clorox treatment of the collected eggs. We introduced additional clarifications in Methods.

Line 350. I lost track of what this entire section is for. Why is it interesting whether they show in vitro activity? What is the take-home message for this whole section?

Demonstration of in vitro activity was necessary because some N4C and N6A MTases were previously reported to display overlapping target base specificities (Jeltsch et al. 1999; Jeltsch 2001), and amino-MTases in general show polyphyletic origin (Bujnicki 1999). It was reassuring to see only 4mC addition and not 6mA addition by N4CMT, indicating that it acts exclusively as N4C-MTase and is unable to add 6mA marks. The most interesting finding, however, was the identification of preferred substrates, defined by the intrinsic ability of the chromodomain-less version to recognize certain motifs in DNA which confer substrate recognition properties to previously unmethylatable substrates, highlighting the dual recognition mode of the enzyme (the regional mode conferred by the chromodomain, and the sequence-specific mode likely inherited from bacteria via the TRD). We have emphasized these points in the text, while delegating the lengthy description of our search for minimal substrates to the Supplementary Note 2 and Supplementary Fig. 7.

Curious that both bdelloids and monogononts have lost the C5-MTases, as well as nematodes and tardigrades. What is the hypothesis for why they lost that?

The evolutionary forces that underlie loss of DNA methylation remain unclear, and such losses have occurred independently in multiple branches on the tree of life. While Drosophilids and rhabditid nematodes are the best-studied case of C5-MT loss, recent observations also include myxozoan cnidarians (Kyger et al 2020). Other interesting examples include loss of de novo but not maintenance C5-MTases, as in Cryptococcus (Catania et al. 2020). It is hard to discern a single underlying reason, as it is more likely that lineage-specific circumstances would be involved. In general, DNA modification is secondary to histone modification in the epigenetic hierarchy, and lineage-specific compensatory changes are expected to evolve in each case. We would prefer to leave this discussion out of the scope of the present work, as it would be purely hypothetical.

Reviewer #3 (Remarks to the Author):

In this work, Rodriguez and colleagues aim to demonstrate that bdelloid rotifers present N4-methylcytosine in the genome (4mC). To test this, they identify a putative 4mC methyltransferase that has no close homology to eukaryotic sequences. Then perform DIP-seq (antibody-based immunoprecipitation followed by sequencing) and use PacBio reads to identify 4mC and 6mA in the genomes of two very closely related “species” of *Adineta vaga* rotifers. Then express the putative methyltransferases in vitro and in bacteria to verify the methyltransferase activity. Additionally, the authors perform ChIP-seq of three histone-tail modifications (H3K4me3/H3K9me3/H3K27me3) to understand the possible crosstalk with 4mC (and 6mA). Finally, they try to identify a putative reader of 4mC in the *Adineta* genome, and perform in vitro experiments to test the capacity of a Methyl Binding Domain protein to perform this function.

Overall, it would be an exciting finding to show that 4mC is present in rotifer genomes. This manuscript presents a compelling case for this being the case, but offers quite weak support for the genomic distribution of 4mC in the genomes of these species or its possible functions. Then, many of the other claims raised across the manuscript are also weak. The manuscript is unfocused and too long, difficult to follow in many sections, and should be shortened drastically, with some sections dropped. Also, the methods used to analyse a lot of the data in this manuscript are not aligned with the standards of the field. In sum, I would not recommend the publication of this manuscript in the current form in *Nature Communications*.

We thank the reviewer for thorough analysis of our data, which helped significantly in improving the manuscript. As suggested, we have drastically shortened the text, creating four Supplementary Notes; aligned the analysis methods with the prevailing standards; and strengthened the evidence for non-random 4mC genomic distribution, as is expected from the presence of the chromodomain in N4CMT. We hope that our responses to the points below help to strengthen our case.

The first criticism in terms of novelty is that *Marchantia polymorpha*, a land plant, has been reported to contain 4mC recently (<https://www.biorxiv.org/content/10.1101/2021.02.12.428880v1>). In the *Marchantia* work, the authors identify the responsible 4mC methyl-transferases and are able to profile 4mC by using whole genome bisulfite sequencing and SMRT-seq, to achieve precise base-resolution identification of 4mC sites (unlike in this manuscript). That work does not

imply the findings in rotifers are not meaningful and potentially very novel, but the authors should discuss that work throughout the manuscript and avoiding claims such as this being the first example of 4mC in a eukaryotic genome.

We adjusted the abstract and the text, focusing mainly on the epigenetic aspects of 4mC mark, as already emphasized in the title. We now discuss the Marchantia work, noting that the enzyme lacks additional N- or C-terminal domains and therefore adds 4mC marks indiscriminately across over one-half of all CpG sites in this genome. We removed the “first” claims, however please note with regard to priority that the time stamp for our work in Research Square is March 2021: www.researchgate.net/publication/353941577_Bacterial_N4-methylcytosine_as_an_epigenetic_mark_in_eukaryotic_DNA as we tried to obtain “Under review” status for our preprint.

In general, I believe that the methyltransferases identified in this manuscript are possibly active in rotifers, and they are likely originated from bacterial LGT (although the sister group in the tree are cyanophages, which supports a viral origin). But the distribution of 4mC (and 6mA) is way less clear, making the function of these modifications dubious in rotifers. DIP-seq is well known to be prone to artifacts, and the lack of coherence between the DIP-seq datasets between the two genomes used in this manuscript clearly shows that.

We employed DIP-seq primarily for initial evaluation of genome-wide distribution, being conscious of its disadvantages. As mentioned in the text, DIP-seq is a low-resolution methodology which limits the power of correlation analyses to the length of DNA fragments used for antibody binding (250-450 bp), not to mention residual IgG binding to non-modified fragments inherent to the method (original lines 220 and 488). Further, the DIP-seq datasets may be expected to overlap only partially for biological reasons, as we employed two distinct morphospecies, with 88% genome identity between Av-ref (clonally maintained under laboratory conditions for over 30 years) and the natural isolate AvL1 recently captured in the wild (line 124). We assumed that the lab strain may have gradually lost some of its methylation potential due to the lack of selective pressures experienced by the field populations, and indeed one of Av-ref N4CMT alleles shows lower activity than the other, differing by 6 aa substitutions. Nevertheless, after changing the visual peak representation method to DeepTools2, we are observing better coherence between two DIP-seq datasets (cf. Fig. 2a and Supp. Fig. 6a). The elevated 6mA signal from the gene set in AvL1 can be partially explained by the presence of unknown TE types in the gene set, which are erroneously annotated as genes by automated annotation pipelines but escape detection by repeat-mining tools due to the unusually low TE copy numbers in bdelloids (3-4% overall TE content, with most TE families containing only a few copies). Genometric correlation methods (original line 189, Table S4; added Supplementary Note 1 and Supplementary Fig.4) also detect positive correlation between 4mC and TE annotations in both genomes, while 4mC and gene annotations show negative correlation.

Also, the analysis of “peak coverage” used in the figures is very unusual, this type of data, like ChIP-seq, is shown as relative enrichment versus input (IgG control), or directly as normalised read coverage.

We agree that the term “peak coverage” may confuse the reader and have changed it to “IP occupancy” as in Fu et al. 2015 Fig. 3A (original ref #73), explaining the Y-axis in more detail in the Methods. Initially, DIP-seq profiles were obtained with `annotatePeaks.pl` from the Homer suite (Heinz et al. 2010), a program for performing peak annotation and associating ChIP-seq with nearby genes (or TEs) or expression data, which is actively maintained by the Benner Lab at UCSD. Unlike `deepTools` with graphical output, `annotatePeaks.pl` prints an output table with the “ChIP-Fragment Coverage, which is calculated by extending tags by their estimated ChIP-fragment length” (<http://homer.ucsd.edu/homer/ngs/quantification.html>) values, i.e. the ChIP-depth per annotation along coordinates, which we then post-processed for graphical output. We have now switched to `deepTools` suite to improve analysis and graphical representation and are plotting DIP-seq read occupancy instead of peak occupancy.

Then, all the SMRT-seq data is also shown as “coverage”, when methylation data across the literature is always shown as a fractional value. Methylation goes from 0 to 1 on a given base depending on the proportion of reads that support methylation, and that is what is informative when showing “metaprofiles” of any methylation mark. This is not done in this manuscript, and renders it very difficult to interpret.

We opted for presenting the methylation fraction (from 0 to 1) and linear representation (along contigs) in circos plots, which is represented as height (y-axis) in the corresponding methylation layer for 4mC and 6mA along with PacBio read coverage (Fig. 4h and Supplementary Fig. 6), as in Liang et al. 2018 Figure 2B and Figure S4C (former ref #74). In Fig. 2e, we present the methylation fraction distribution at modified sites detected by SMRT-seq, and note that most of the 4mC sites are nearly fully methylated, as is also visible on Circos plots. On “profile” plots with SMRT-seq methylation (Fig. 2c) we employed the same approach as in Beh et al. 2019 (ref #72 with Fig. 3C representing “Number of 6mA sites”), and Liang et al. 2018 (ref #74 with Fig. 3D,F representing “6mA occupancy”). Both methods used the distribution of methylated sites,

employing different bin sizes for plotting purposes. We have removed the confusing term “coverage” and plotted 4mC and 6mA occupancy near the 5’ TE insertion boundary in the revised Fig. 2c using two distinct windows and bin values, with the y-axis now named “SMRT 4mC/6mA occupancy”.

Also, the methods used by the authors do not show coherence between DIP-seq and SMRT-seq, which makes the whole claim that these marks are enriched on transposable elements dubious.

Indisputably, the DIP-seq methodology is prone to sensitivity issues, nevertheless the overlap that we find between DIP-seq and SMRT-seq modification patterns is considerable, with 36% of 4mC DIP-seq peaks and 32% of 6mA peaks overlapping with 4mC and 6mA identified by SMRT analysis, respectively. Given the modest percentage of modified bases, with 0.0643% of the total cytosines in the assembly (21,016 4mC modifications) and 0.0236% of total adenines (17,886 6mA modifications) defined by SMRT-seq, the overlap between DIP-seq and SMRT-seq is quite substantial, considering the likely sources of natural variability such as the developmental stage (animals vs eggs). We further provide several independent lines of evidence for enrichment at TEs and tandem repeats, collectively supporting our claims. Concentration of modified bases over transcriptionally active TEs is readily visible in the Circos plots of Supplementary Fig. 6c,d. Furthermore, manual inspection of 36 unannotated high-density 4mC regions (originally described in Supplementary Note 3 and shown in Supplementary Fig. 6f and Supplementary Table 12) revealed that at least one-half corresponds to unrecognized TEs, further strengthening our claims. We have now inserted this information from the supplement into the main text.

Including the 6mA data is also quite problematic. The presence of 6mA is highly contested in animal genomes, and the data presented here is not robust. 6mA could be RNA contaminations (as found in mammalian species). The claims regarding this section do not offer much novelty and remain contentious.

We are aware of the controversy surrounding 6mA, especially in mammals, and have exercised caution in interpreting 6mA data, most of which was delegated to the supplement. Notably, the most represented motifs at 6mA addition sites (Fig. 2d) do not match the RRACH motif characteristic for RNA MTases, which is one of the criteria used by Douvlatianotis et al. (2020). Another line of evidence against RNA contaminations as the main source of 6mA, which we also present in the supplement, is the occurrence of the 6mA double peak in the non-transcribed region upstream of the TSS in a set of gene orthologs from two species, which is visible not only in SMRT-seq, but even in the low-resolution DIP-seq data from Av-ref and A. vaga. We suggested METTL4 as the most likely 6mA-adding enzyme, due to its N-terminal domain which has the potential to interacting with DNA and to the presence of a nuclear localization signal, although its activity still needs to be tested biochemically. We cannot rule out that some 6mA sites closely linked to 4mC in SMRT-seq data might be caused by trickle-down of IPD signal from 4mC (similarly to 5mC), although this is unlikely to affect antibody specificity in DIP-seq, and a substantial fraction of 6mA sites is distanced from 4mC (as is visible in circos plots Fig. 4h and Supplementary Fig. 6). Considering multiple publications on 6mA, especially in non-mammals (examples in Supplementary Table 1, to which we added a reference emphasizing low reliability of mammalian data), we believe that, despite its controversial nature which is acknowledged, 6mA may still be of biological relevance in rotifers, which borrowed multiple genomic features from fungi. As our MS is not focused on 6mA, keeping the 6mA data as a supplemental addition to our principal findings, without placing much emphasis on it in the main text, should make it available to other researchers for independent evaluation.

Would be good to show the intron-exon structure of the gene with RNA-seq data, and presenting some extra quality checks ensuring these genes are not found in bacterial scaffolds and are indeed encoded as host genes.

We now present the intron-exon structure in a new Supplementary Fig. 1, which also shows RNA-seq coverage for two allelic variants (a) and the syntenicity of genomic environments in the genus Adineta (b). An additional Supplementary Fig. 2 shows intron positions in the amino acid sequence alignment of the N4CMT proteins from ten bdelloid species, as well as the C-terminal fusion to the chromodomain and distinct N- and C-terminal extensions clearly distinguishing these proteins from shorter bacterial counterparts. These four lines of evidence (two alleles, introns, preservation of syntenic environment on long contigs over evolutionary time scales, fusion to a eukaryotic domain) combine to yield irrefutable evidence of N4CMT presence in metazoan hosts. A bacterial N4N6-MTase-containing contig (OESY010524654) was bioinformatically identified as a contaminant in Rotaria magnacalcarata assembly and is mentioned in Methods as an example.

Since both genomes are so close, one way to make the claims more robust would be to show orthologous regions across both species with similar profiles in terms of SMRT-seq base modification and DIP-seq.

The difficulty in showing orthologous regions is the highly variable nature of TE insertion sites, which do not coincide between genomes even if they are closely related. Thus, we chose to focus on treating TE features as a whole or grouped by length, and patterns for representative TE insertions are shown on the circos plots in the supplementary Fig. 6. Ortholog

analysis was employed for 6mA and is presented in the Supplementary Note 3 and Supplementary Fig. 9 for orthologous gene subsets in both genomes (Av-ref and AvL1) with similar profiles for SMRT-seq, DIP-seq and RNA-seq. The small increase of 6mA initially observed in AvL1 by SMRT-seq near the TSS (Supplementary Fig. 9f, right panel) took us to explore DIP-seq profiles those genes (1212 gene models) and, after blastp search, on their Av-ref homologous genes. Confirming our AvL1 observations, Av-ref homolog gene models not only showed similar DIP-seq peak profiles (Supplementary Fig. 9i), but also significantly higher RNA-seq transcript values than average (Supplementary Fig. 9j).

Figure 1f contains some errors. There is no need to establish “Phylum (Class)”. This is not very useful, plus, the authors do not stick to this classification for many groups. E.g. Fungi are not a Phylum. “Protist” is not a valid category, Amoebozoans are more closely related to Animals and Fungi than to Ciliates. Similarly, Ciliates should be sister group to the plant lineages. “Higher plants” does not have any meaning, there is no “lower” or “higher” plants, please refer to the correct group (Embryophytes?). Early Metazoa -> Non bilaterians. Green algae -> Chlorophyta? Some of the common names are confusing.

The taxonomic issues in Fig. 1f have been corrected. We removed “Phylum (Class)” designation and used more appropriate designations for each group, as suggested. Amoebozoa were moved up with other opisthokonts. Ciliates were designated as SAR instead of protists. Higher plants were replaced with Tracheophyta (vascular plants). Green algae included Chlorophyta and Streptophyta minus Embryophyta, as explained in the legend. Taxon inclusion is necessarily limited by availability of sequenced genomes available for BLAST searches at NCBI.

Figure 2 is a mess. Why there are 2 letter codes? Some of them are in capital, others in lowercase, very confusing.

We apologize for the mishap, which occurred during reformatting. All letters are now in lowercase.

Figure 2a. What does “coverage” represent? Peaks? What is “peak” coverage? Number of peaks that intersect with that window? That is not how these metaprofile plots are usually computed. These take the aligned reads from the DIP-seq (or ChIP-seq) and transform it to a read coverage plot, corrected by coverage (e.g. Counts per Million), and, if available, input DNA (IgG only). Those are the values then used to compute “coverage” plots such as these.

Please see above for explanation for the initial use of the confusing term “coverage”. Initial DIP-seq plots for peak occupancy have now been replaced with DIP-seq read occupancy plots using deepTools2 (Fig. 2a; Supp. Fig. 5), and “coverage” was replaced with “IP occupancy”, as explained in detail in the legend and Methods.

The authors have used DeepTools in Extended Data Fig 2 c/d, but that figure is rather unclear. First, it shows enrichment in all features for both 4mC/5mC. It also says “Relative Fold Enrichment” (respect to what?). TS should be TSS (Transcriptional Start Site), and TTS should be TTS (Transcriptional Termination Site), these are standards in the field, why change them? DeepTools plots can be modified in Inkscape or Illustrator, that would be highly beneficial, since “Gene distance” is not very intuitive when representing Transposable Elements.

Extended Data Fig 2 c/d showed enrichment mostly on TEs, for both 4mC and 6mA (DIP-seq). We made use of deepTools cluster analysis (where the heatmap matrix is split into clusters using the k-means algorithm) to sort features by DIP-seq coverage and display different profiles contributing to the initial plot. This is how we identify a group of TE annotations (cluster 1 & 2 in Supplementary Fig. 5d, right profiles) with an enrichment of DIP-seq 4mC marks, and no such enrichment for 6mA. To our understanding, deepTools “relative fold enrichment” is calculated as the number of reads overlapping each feature from a given BED/GTF file, after initial normalization by RPGC (reads per genome coverage). We have restored the TSS/TTS designations, which were initially changed to TS/TT to avoid font overlapping with the window size in heatmaps.

The Figure 2a “coverage” metaprofiles do not fit at all with those shown in Extended Data Fig 2a/b. This highlights how unreliable are these DIP-seq techniques, which show very different results in two very closely related rotifers.

As recommended, we switched to deepTools2 in figures, and abandoned other graphical representations of the peak calling outputs, keeping only the number of MACS peaks in the text. DIP-seq data represent one out of several lines of evidence, as we are aware of low reliability of each individual method. Some difference in results from two rotifers is anticipated, because the Av-ref strain has been maintained in the lab for over 30 years and has not experienced the same selection pressures as the AvL1 natural isolate (see above). We achieved further improvement in DIP-seq data by additional clean-up of unannotated TEs, which unavoidably penetrate the outputs of the automated gene annotation pipelines. As a result, we could improve the quality of the figures to look more similar. However, the generally low TE copy number prevents a complete clean-up of unknown TE types, and thus some entries in the gene set may represent yet unidentifiable TEs.

Figure 2d. I believe the SMRT analysis software provides motifs enriched on methylated bases, why did the authors show MEME-ChIP which clearly does not fit this data?

*SMRT analysis software contains the motif analysis tool “MotifMaker” which is a motif finding algorithm displaying similarities and differences with other motif finders such as MEME-ChIP. The reason we did not employ this additional SMRT analysis module was because Motif Maker is described in the PacBio documentation and white paper as “a tool to identify motifs associated with DNA modifications in prokaryotic genomes” (<https://github.com/PacificBiosciences/MotifMaker>), which discouraged us from using it in a eukaryotic genome. MotifMaker is better suited for finding a single motif, while others like MEME-ChIP are based on aligning motifs and identification of a major variety of motifs. Single motifs could be more representative for prokaryotic genomes, like GATC/CTGCAG in *E. coli* (Fang et al. 2012, Genome-wide mapping of methylated adenine residues in pathogenic *Escherichia coli* using single-molecule real-time sequencing, *Nat Biotechnol*) which are useful for identifying highly specific targets for methylases. In eukaryotes, at least for 6mA, methylation events seem to be less of a “single” motif due to other functional regulations (Fu et al. 2015, ref #73; Wu et al. 2016, ref #84). We followed the steps outlined in Greer et al. 2015 (ref #21) and Liang et al. 2018 (ref #74) for motif identification, by extracting the sequences upstream and downstream from each methylated position, and used MEME-ChIP because it would provide a more representative mixture of motifs, better reflecting the complexity of regulation in eukaryotic genomes.*

Figure 2g and h. PacBio counts should not be used, methylation levels are always shown as fractional levels since this is a technique with base resolution.

In Figure 2g and h we intended to represent the distribution of methylated sites (SMRT-seq) across the genome and to make comparisons between the number of observed 4mC/6mA sites in gene bodies as well as in transposons and tandem repeat regions. Similar distribution graphs are shown in Liang et al. 2018 (ref #74) in Figure 3A/B, which counted the number of 6mA sites (from SMRT-seq) in intergenic region and genes. We were mostly following approaches for 6mA characterization, as we needed to characterize only 6mA and 4mC, but not 5mC.

Figure 2i indicates that *Adineta* has enriched 4mC and 6mA on transposable elements, when other plots in this same figure indicate that 6mA is not enriched on these elements. This is very confusing. Also, showing “dynamite plots” is bad practice, these values should be shown as a distribution (boxplot?). I still do not know why SMRT counts are used instead of fractional values.

*While browsing (Gbrowse or IGV) through the *Adineta* genome, we noticed the dual enrichment of 4mC and 6mA, which was also notable when looking at SMRT counts per annotation (Fig. 2g). However, we observed that the distribution was slightly different: while 6mA sites are rather randomly distributed over the TE body or upstream/downstream, 4mC shows a more precise distribution towards the 5' insertion point. This is visible in the plot of Fig. 2c using two different window/bin sizes. We further noted that some TE copies are more prone to carry methylation marks than others. This is why we divided the TE dataset into three categories (full, medium, short) based on the length of a copy compared with the reference TE, i.e. assuming that more recently acquired TEs would have a more intact sequence (full/medium) vs old truncated ones (short), indicating that active TE copies are preferentially targeted. The distribution of the data is strongly skewed towards zero methylation (left-skewed), which is likely due to TE under-annotation (see above) and transcriptional inactivity of many TEs in the *Adineta* genome. This makes it difficult to represent the data in a boxplot for standard distribution. Our intention was to illustrate the average number of modifications using the graph in Fig 2i, so that we could make comparisons between TE subsets without browsing through every TE annotation or determining transcriptional activity of individual copies, which is not feasible with this type of data. Nevertheless, in a few individual examples which are provided in Supplementary Fig. 6, the over-representation of methyl marks over annotated full-length or nearly full-length TEs is readily visible.*

Figure 2c. What is “coverage”? PacBio base modification relies on coverage on positions, but then gives fractional values on any given position (0 to 1 values). In fact, “coverage” is a meaningless feature to determine the level of a base modification in any given position, but the “fraction methylation levels” should be averaged and displayed for a metaprofile like this. Again, I would recommend the authors to clarify what this is and maybe use DeepTools2. Why does it say 2.5 Kb window? The plot shows 3kb. Same for the one below, 500 bp window but there are 600 bp shown in the x axis.

“Coverage” was replaced with “SMRT 4mC/6mA occupancy”. We have edited the text to make it clear. The plots were adjusted to show the correct window size of 2.5 kb (top) and 500 bp (bottom).

Extended Data Fig 3 is very hard to read, pixelated when zoomed in. There is not need to show these are “Circos” plots, simple Genome Browser representations would be much easier to interpret, zooming out or in depending on the features that need to be highlighted.

We have reorganized this figure, removing non-informative parts from four circles, reducing pixelation, and adding ID's for each track for easier interpretation. We are using browser snapshots in the new Supp. Fig. 1 to demonstrate intron-exon structure and expression, but Av-ref and AvL1 data poorly fit onto the same browser due to 12% divergence of AvL1 from the reference. Circular plots have an added advantage of combining several genomic regions within the same panel, avoiding the pileup of multiple plots as in Supp. Fig. 1.

The sequences of the conserved motifs shown in Figure 3f do not contain any CG dinucleotide, which in figure 2b was shown to be the preferred substrate of these enzymes according to SMRT-seq data. This is inconsistent and is not discussed.

Our explanation of the conserved motif significance apparently was not clear enough – in vivo methylation sites at CG dinucleotides within the 460-bp repeat were shown with red arrows in Fig. 3g (now Supp. Fig. S7e), however no arrows are marked within the conserved motifs, which led us to hypothesize that these motifs ensure sequence-specific target recognition by the MT moiety independently of the chromodomain, as confirmed by efficient in vitro recognition by the chromodomain-less N4CMT mutant, but do not necessarily coincide with methylated sites in vivo, where chromodomain-based recognition would also play a role. We now make this distinction clearer in the text.

How are the + and – in Figure 3g determined? With an immuno-dot blot? Where is it? Not all combinations seem to be found in h and i panels. H and I panels are hard to interpret and poorly explained in the figure legend.

The presence or absence of 4mC mark after treatment with N4CMT, labeled as “+” or “-”, was determined based on immuno-dot blot experiments with anti-4mC antibody. Each DNA was tested at least two times in independent experiments, and one of the respective dot blots is included in updated Supplementary Fig. 7. In each experiment we included control spots: reaction mix without DNA template and mix with DNA but without N4CMT, to control for background noise and non-specific chemiluminescence signals. We have re-organized Fig. 3 and moved four panels, including h and i, into the Supplementary Fig. 7 as part of the manuscript shortening. The immuno-dot-blots covering all combinations of + and - are now combined in the supplement. Explanations in the figure legends have been improved.

Figure 4. H3K9me3 and H3K27me3 are not found usually on the same sites, one is constitutive chromatin and the other facultative (at least in mammals). This “expectation” is not very realistic. To validate the CHIP-seq, it would be good that the plots in Figure 4a/c would be sorted the same way, e.g. using the one to one orthology relationships between assemblies/species. Also, it would be good to add more information on the figure itself, reading the legend is confusing.

Indeed, for a long time the separation of function between constitutive chromatin marked by H3K9me3 and silencing TEs, and facultative H3K27me3 silencing genes, has been the prevailing paradigm. However, there is a growing body of evidence from less traditional models, such as ciliates and bryophytes (recently reviewed by Délérís et al. 2021, PMID: 34210514), that H3K27me3 can serve as an ancestral TE-silencing mark, with the subdivision into constitutive K9 and facultative K27 occurring later in evolution, in higher plants and mammals. Although we have employed orthology to assess the significance of the genic 6mA marks (supplementary note 3), it would be problematic to employ it for validation of TE marks since TE insertions lack orthology and can be analyzed only as individual insertions, or in some combinations (e.g. full-length vs truncated). The anti-H3K9me3 and anti-H3K27me3 antibodies have been validated for the lack of cross-reactivity, as explained in Methods.

Figure 4e/f. This plot is suspicious, the shape of the “peak” indicates that there is unclear smoothing within that window. Usually peak metaprofiles are centred around the peak summit (or centre), without setting hard borders as if they were genes, which tends to generate artefactual shapes as those shown here. Also, figure legends are barely visible.

Figures 4e/f have been corrected by centering on the peak summit. The font size in the legends has been increased.

Figure 4h. Almost impossible to read, too pixelated. Better to show linear genome browser snapshots highlighting some associations instead of dumping too much data in a circular plot that is difficult to follow.

We removed the least informative regions from the circular plot, which helped to improve resolution and reduce pixelation. As explained above, genome browser snapshots are difficult to use with divergent reference genomes and occupy a lot of

space, as may be seen from the new Supplementary Fig. S1, while the circular plots accommodate several contigs showing associations between DIP-seq, SMRT-seq, H3Kme, RNA-seq, and small RNA layers.

Figure 4g. This should resemble figure 4f using a metaprofile displaying the fractional level of methylation (4mC/6mA) on those peaks, and not a “dynamite plot” that is difficult to interpret.

Unlike Fig. 2c, which shows an enrichment of 4mC towards the 5' boundaries of TEs, the H3K9me3 and H3K27me3 histone marks do not show a specific profile with regard to DNA methylation. As visualized in Fig. 4h and Supplementary Fig. 6, we normally see several peaks (both H3K9 and H3K27) on a contig region which shows 4mC and 6mA DNA marks distributed quite broadly. The H3K4 mark, as expected, is commonly found in gene-rich regions. We opted to represent in this plot the average number of modifications for each histone mark, as a proxy to better visualize global patterns across the genome. A large number of annotations with zero counts presents a skewed distribution and is difficult to eliminate.

The section on the small RNAs feels confusing and unnecessary for this manuscript.

We have moved this section to the Supplementary Note 4. It would be undesirable to delete it altogether, as it strengthens the link to TE transcriptional activity, which serves as a prerequisite for small RNA production (as is well known from the literature), as well as for 4mC deposition (as may be inferred from Fig. 2i).

The section about SETDB1 as a possible reader of 4mC is also very weak and unnecessary. Not all animal MBD proteins bind to methylated cytosines, that is well established. MBD1/2/3/4 and MeCP2 are the families that have been shown to recognise methylated CpGs in other animals. In some cases, like *Drosophila melanogaster*, it harbours a highly divergent MBD2/3, that has been shown not to bind to preferentially to methylated CpGs, which is expected since *Drosophila* lacks 5mC methylation. The competition experiments shown in Figure 6 are not very strong indication of this domain preferentially binding to 4mC methylation.

We fully agree that MBD is a generic DNA-binding domain, and this was clearly stated in the main text, with a full overview of MBD proteins presented in supplementary Fig. 11. While the difference in binding affinity of the Av314 variant is not drastic, it is nevertheless significant, and we provide the source data file containing the results from replicate experiments. Other observations in support of Av314 role are (i) the lack of detectable differences in DNA binding for the other five paralogs, and (ii) identification of Av314 as the only variant for which the loss is associated with increase in vertically-transmitted LINE-like TEs (Fig. 6b,c). Collectively, these lines of evidence fit well with the “reader” concept, as the highly unusual amplification observed for SETDB1 could provide the raw evolutionary material for developing 4mC preference and serve as the missing link to H3K9me3 as implied by our initial finding of the chromodomain in N4CMT. Although this section could be moved to the supplement, we feel that there is no overstatement in keeping it in the main text with full explanation of the caveats, however removing it altogether would leave a void.

Figure 6f. Given the data shown in this manuscript, it feels premature to draw a model on how this system works in tardigrades.

We explicitly say that the model in rotifers is hypothetical, nevertheless it is fully consistent with the data presented in the manuscript and assists the reader in placing our findings into the plausible chromatin context, in agreement with known functional properties of the CMT and KMT enzymes involved. These are the only connections that were drawn in non-dashed lines in the model. We also left out other known processes that may be involved but are less relevant to the present work, such as deacetylation, ubiquitination, sumoylation etc.

Claims such as “Finally, it demonstrates that horizontally transferred genes, contrary to the established view”. Lateral Gene Transfer is contested as a major source of innovation in eukaryotic lineages by some authors, but this is far from being “the established view”.

We changed the wording in this sentence: “Finally, it demonstrates that horizontal gene transfer, the role of which in eukaryotic regulatory evolution is a subject of intense debate...”. The emphasis here is on regulatory evolution, as opposed to the much more widespread HGT of “operational” genes controlling specific metabolic reactions.

Plots such as those in Extended Data Fig 5 e,f,g,i are basically noise. This is likely due to the non-standard ways of representing this data, and no information can be extracted from these.

We agree that most of the genic 6mA marks in this figure are noise, except for two peaks visible upstream of the TSS, which are the only meaningful locations, as they appear independently in two orthologous gene subsets from Av-ref and AvL1. The relevant information that can be extracted from these peaks is their localization in non-transcribed regions, which argues

against RNA origin of the signal and justifies keeping it as supplementary material. The Y scale in (e) has been adjusted to that in (f), emphasizing the noisy character of the genic marks which are identifiable only in a subset at the peak location.

Reviewer #4 (Remarks to the Author):

Comments to the author

Rodriguez et al. performed an impressive research on 4mC occurrence in eukaryotic DNA. The major result underlines the 4mC presence in bdelloid rotifers by combining multiple trials I believe the results reported would be interesting to wide spectra of biologist and other specialists.

We thank the reviewer for the positive evaluation of our work.

Points raised:

1. In figure 1a,1b, is it more intuitive to mark the direction with the N-terminal or C-terminal instead of using numbers to mark the direction? In addition, using cylinders to roughly represent multiple enzymes seems insufficient. Is there a way to show the structural differences more clearly? I'm not whetted on it, but think the article would be improved, if the authors consider this and take some actions.

We have added Supplementary Fig. 2 to show the alignment of full-length ORFs indicating the length of each protein, functional domains and motifs, catalytic residues, secondary structure elements, and the degree of conservation. Since each species shows the same structural organization, the approximate schematic representation in Fig. 1a appears adequate, as it reflects only the basic domain architecture and the cylindrical MTase domains are evolutionarily related. Using N and C to mark the direction does not seem more informative than indicating length.

2. The author mentioned that for gene profiles, the modification density is much lower, and it seems the opposite compared to TE profile. Can the authors explain the opposite distribution phenomenon of gene profile and TE profile in a more understandable way?

We have redrawn the plots for gene profiles and TE profiles for both Av-ref and AvL1 (Fig. 2a; Supplementary Fig. 5a), and the increased density of modified bases over TEs is visible in both, apparently due to preferential targeting of N4CMT to silent H3K9/27me3-marked chromatin via its chromodomain moiety. Modifications in genic regions are much less pronounced and do not show robust association using statistical correlations (Supplementary Table 4). We have explained this in the text and introduced an additional Supplementary Note 1 and Supplementary Fig. 4 to clarify the details of statistical analysis.

3. As far as I know, coverage level is a major effect factor on correction efficiency. In the range of 10x to 50x, the correction result increases as the coverage increases (https://www.uni-wuerzburg.de/fileadmin/07030400/AG_Genomics/Proovread/proovread-preprint.pdf). Therefore, does the author think that after increasing the coverage level, the overlap between SMRT-seq and DIP-seq is higher?

Indeed, as we document in Supplementary Table 6, the number of modified bases increases with coverage, and further increase of PacBio SMRT-seq coverage could shed more light onto regions with undetected methylated sites having lower SMRT coverage profiles, e.g. towards telomeres. Nevertheless, most discrepancies in the overlap between the two techniques may be attributed to DIP-seq, where the methodology is highly prone to sensitivity issues, notwithstanding the difference in developmental stage explained above. However, given the modest percentage of modified bases, with 0.0643% of the total cytosines in the assembly (21,016 4mC modifications) and 0.0236% of total adenines (17,886 6mA modifications) as defined by SMRT-seq, the overlap between DIP-seq and SMRT-seq is quite substantial, with 36% of 4mC DIP-seq peaks and 32% of 6mA peaks overlapping with 4mC and 6mA identified by SMRT analysis, respectively.

4. In line 247, the author mentioned "At 4mC sites, CpG and CpA dinucleotides are the most prevalent, making up 74% of modified doublets." Please explain the calculation process.

We have updated the methods section with an explanation. Briefly, the upstream and downstream 10-bp sequences from 4mC and 6mA modification sites were extracted for motif identification, and the adjacent nucleotide base of the methylated sites was pulled out for counting the proportion of doublets. This function is provided in the SMRT portal.

5. In line 259, the author reports the different methylation levels of 4mC and 6mA. Whether the conclusion drawn is related to the state of the sample. In other words, the methylation levels show differences at different time nodes.

We consider the differences in methylation fraction to be mostly due to inherent characteristics of each modification, with 4mC being more reproducible and appearing in a much higher fraction of sites, and 6mA more dynamic and subject to developmental stage- and tissue-specific differences. Our current goal was to discern the most stable modification patterns which would be largely preserved across tissues and developmental stages, and while there may be some interesting developmental dynamics of methylation patterns, we considered such studies to be outside the scope of the present work.

6. Please use the definite article accurately and polish other details that I haven't noticed. For example, "the genome-wide", "We visualized the distribution", etc.

Corrected.

7. The ordinate of Fig. 2B is not clear, and the picture resolution needs to be improved.

The Y axis has been redrawn and the resolution improved.

8. It may be due to the typesetting that makes Figure 2 look confusing. Please adjust the position and label of the picture to make it easier for readers to read. In addition, m4C, m6A in the 2F should be modified to 4mC, 6mA, please keep the abbreviations consistent.

The figures have been corrected and the labeling has been improved. Abbreviations have been made consistent.

9. Some typos should be revised.

Corrected.

REVIEWER COMMENTS

Reviewer #2 (Remarks to the Author):

Thank you for making changes to address my comments.

Reviewer #3 (Remarks to the Author):

Despite the authors have done a great deal to improve the manuscript, several of my complains have not yet been addressed.

First, the *Marchantia* work is simply dismissed. Adding it as reference 87 of the Supplementary Material does not work. I don't know why the authors do this, since the *Marchantia* results increase the value of their own findings and puts it into context way better than trying to ignore it, maybe just to be "the first" to report 4mC in an eukaryote. I really don't think it is important who posted this first, both works are now available to read by anybody, so ignoring the preprint makes no sense. Also, when it is a case of 4mC usage in a completely different branch of eukaryotes. If the reports of *Adineta* 4mC are robust, it would be important enough, the first case in animals, with a LGT methyltransferase, etc. In fact, the title could well be "Bacterial N4-methylcytosine as an epigenetic mark in metazoan DNA" or "rotifers", which is more precise and impactful enough.

Then, the analysis of the data.

- 1) The authors seem to have taken just the DIP-seq read coverage with DeepTools and plotted it without correcting it with the IgG control. If using DeepTools, it would be important to use `bamCompare` (e.g. `bamCompare -b1 DIPseq.bam -b2 IgGcontrol.bam -o log2ratio.bw`) to create the bigwig file and plotting those corrected values instead of raw coverage. The same way MACS2 uses the IgG control as background to know what a real peak is, as readers we would like to see this data corrected for noise. For instance, the plots showing exactly the same profiles of 6mA and 4mC in genes for *AvL1* (Supp Fig S5) could well be that the sequence bias is driving these patterns irrespectively of true signal. Same thing goes for Figure 4e/f.
- 2) In terms of showing the fractional data in Circos plots. Those are very difficult to read. For instance, Figure 4h. There are two tracks, called "Illumina 1" and "Illumina 2" (what is that anyway?), but then the 4mC and 6mA values appear on those tracks... I guess this is the result of wanting to put both assemblies on the same figure, but it is very hard to read/interpret, moreover when different assemblies have different data types. Also, none of those plots include a y-axis. What are those fractions? 0 to 1? 0 to 0.02? The authors say that it is usually 100%, but when I ask for a IGV plot is to clearly see the raw data, and this is not so easily read in the Circos plots. After all, their preferred representation coming from Liang et al 2020 also shows a IGV plot in Figure 1f..
- 3) A clear way to show the overlap between 4mC and the DIP-seq would be to 1- call peaks using the DIP-seq. 2- generate a bigwig file of 4mC (e.g. scaffold position methylation_fraction), maybe just including the CG sites to increase the signal. 3- show the heatmap on those peaks using 4mC DIP-seq (corrected) values (see point 1) and 4mC fractional values together. It is easy in DeepTools to specify different colours / scales for various marks, so this is quite easy to get. That way we would see the co-localization of both techniques on the same plot genome wide, instead of just showing some vague global overlaps. This could be also done for the 6mA, and in fact, in a combination (e.g. overlap DIP-seq peaks).
- 4) The 4mC and 6mA fractional bigwig files should also be used to present metaProfiles as those shown in Figure 2a. The fractional values should also show an average increase of 4mC marks on TEs, and maybe confirm this very subtle enrichment in the TSS.
- 5) The authors say that there are not many TEs in *Adineta*. The heatmaps seem quite data rich. So how many are shown in each plot? That's important to interpret those maps, and to draw potential differences between both strains.

Minor remarks. Abstract – DNA modification is used -> “DNA modifications are used” feels more natural. Then, the second sentence with an “it” is confusing, what is the “it” referring to?

This sentence: “N4CMT adds 4mC to DNA, and its chromodomain shapes the “histone-read-DNA-write” architecture, together with a “DNA-read-histone-write” SETDB1 H3K9me3 histone methyltransferase variant preferentially binding 4mC-DNA, to maintain 4mC and silent chromatin at active transposons and tandem repeats.” Is very long and hard to follow. Furthermore, the connection of SETDB1 with 4mC is very premature to be highlighted like this in the abstract. This would be, at most, a suggestion with the current evidence.

“Our results bring the third base modification into the eukaryotic repertoire” this is simply wrong, there are many other base modifications found in eukaryotes. E.g. J base in Trypanosomatids, hydroxymethylcytosine (and other derivatives), 5-Hydroxymethyluracil in dinoflagellates, or this one (<https://pubmed.ncbi.nlm.nih.gov/31043749/>).

“epigenetic systems to suppress transposon proliferation”. Well, one thing is that it is marking TEs, another one is that is suppressing their proliferation, which this work does not demonstrate.

If 4mC is marking TEs, then it does not have anything to do with “regulatory networks”. Silencing a TE is not a “regulatory” network, but a silencing mechanism, a definition of “regulatory network” according to the Nature publishing group can be found here: <https://www.nature.com/subjects/regulatory-networks#:~:text=Definition,genes in a given genome.>

The authors should define in the introduction what do they mean by “epigenetic”. There’s a lot of controversy in the field, therefore a clear definition of what do they mean is mandatory (since they find this a unique feature of this system).

Reviewer #4 (Remarks to the Author):

Authors have improved the paper. I think the paper can be accepted.

RESPONSE TO REVIEWER COMMENTS

Reviewer #2 (Remarks to the Author):

Thank you for making changes to address my comments.

We thank the reviewer for re-assessing the revised version.

Reviewer #3 (Remarks to the Author):

Despite the authors have done a great deal to improve the manuscript, several of my complains have not yet been addressed.

As detailed below, the current revision addresses each of the remaining comments, which were very helpful for improving data analysis and presentation.

First, the *Marchantia* work is simply dismissed. Adding it as reference 87 of the Supplementary Material does not work. I don't know why the authors do this, since the *Marchantia* results increase the value of their own findings and puts it into context way better than trying to ignore it, maybe just to be "the first" to report 4mC in an eukaryote. I really don't think it is important who posted this first, both works are now available to read by anybody, so ignoring the preprint makes no sense. Also, when it is a case of 4mC usage in a completely different branch of eukaryotes. If the reports of *Adineta* 4mC are robust, it would be important enough, the first case in animals, with a LGT methyltransferase, etc. In fact, the title could well be "Bacterial N4-methylcytosine as an epigenetic mark in metazoan DNA" or "rotifers", which is more precise and impactful enough.

The reference was added in the most appropriate place in the Discussion, which happened to be part of the Supplementary Discussion. We have now moved the corresponding part of the Discussion into the main text (p. 17), with the accompanying reference. It would be ideal if the two papers could be published in parallel to increase the value of each other, however this would be beyond our control.

Then, the analysis of the data.

1) The authors seem to have taken just the DIP-seq read coverage with DeepTools and plotted it without correcting it with the IgG control. If using DeepTools, it would be important to use bamCompare (e.g. `bamCompare -b1 DIPseq.bam -b2 IgGcontrol.bam -o log2ratio.bw`) to create the bigwig file and plotting those corrected values instead of raw coverage. The same way MACS2 uses the IgG control as background to know what a real peak is, as readers we would like to see this data corrected for noise. For instance, the plots showing exactly the same profiles of 6mA and 4mC in genes for AvL1 (Supp Fig S5) could well be that the sequence bias is driving these patterns irrespectively of true signal. Same thing goes for Figure 4e/f.

We thank the reviewer for valuable advice and detailed instructions. For peak calling in ChIP-seq data with MACS2, we used input DNA, rather than IgG, as chromatin control in both strains for background correction. Following reviewer's suggestions, we added the correction step before plotting with deepTools, using bamCompare with both the treatment and the input control, to obtain the log2 ratio between the two. Figure 4e/f has now been updated using these corrected profiles, which satisfyingly led to elimination of the artifactual peak previously appearing in the H3K4 plot. Accordingly, we have removed descriptions of this artifactual peak from the text and legend to Figure 4. In 4mC and 6mA DIP-seq experiments, we used MACS v. 1.4.2, rather than MACS2, for peak calling in DIP-seq data, since it is considered effective in capturing the local genomic sequence biases from a ChIP-seq sample alone, in lieu of the control sample (Zhang et al. 2008). Clustering in Supp. Fig. 5c-d then helps to reveal the minor fraction of the data that is likely to represent noise, displaying no difference between genes and TEs.

2) In terms of showing the fractional data in Circos plots. Those are very difficult to read. For instance, Figure 4h. There are two tracks, called "Illumina 1" and "Illumina 2" (what is that anyway?), but then the 4mC and 6mA values appear on those tracks... I guess this is the result of wanting to put both assemblies on the same figure, but it is very hard to read/interpret, moreover when different assemblies have different data types. Also, none of those plots include a y-axis. What are those fractions? 0 to 1? 0 to 0.02? The authors

say that it is usually 100%, but when I ask for a IGV plot is to clearly see the raw data, and this is not so easily read in the Circos plots. After all, their preferred representation coming from Liang et al 2020 also shows a IGV plot in Figure 1f...

We thank the reviewer for pointing the items which may be difficult to read. We have updated Figure 4h and the legend, and have renamed the confusing "Illumina 1" and "Illumina 2" tracks (representing two different DNA Illumina libraries for Av-ref) to "Illumina 862 bp" and "Illumina 450 bp", in agreement with the actual insert sizes. The two assemblies (strains) were included not only to illustrate the same correlations with methylation for different datasets, but to show similarities/differences of data types (tracks in circos plot) between the two strains. To improve clarity, the strain name (Av-ref or AvL1) is now highlighted at the contigs base in the circos plot. The y-axis scale was added for the PacBio SMRT-seq methylation fraction values (from 0 to 1). Overall, we opted for displaying selected genome sections of the assemblies with circos because, among other things, it can represent large portions of the genome in a single sub-figure within a manuscript, in this case, Figure 4h which represents ~485 Kb of genome data. Although IGV representation and functionality is very useful for routine visualization, depiction of a similarly sized genome fraction as in Figure 4h using multiple tracks for different contigs would require much more space in the main figure. We therefore chose to present selected genomic regions as IGV plots in the supplement (Supplementary Fig. 8).

3) A clear way to show the overlap between 4mC and the DIP-seq would be to 1- call peaks using the DIP-seq. 2- generate a bigwig file of 4mC (e.g. scaffold position methylation_fraction), maybe just including the CG sites to increase the signal. 3- show the heatmap on those peaks using 4mC DIP-seq (corrected) values (see point 1) and 4mC fractional values together. It is easy in DeepTools to specify different colours / scales for various marks, so this is quite easy to get. That way we would see the co-localization of both techniques on the same plot genome wide, instead of just showing some vague global overlaps. This could be also done for the 6mA, and in fact, in a combination (e.g. overlap DIP-seq peaks).

Initially, we did not use the bigwig (.bw) format for PacBio SMRT-seq methylation, since it is meant for dense and continuous datasets. Following the reviewer's suggestion, we explored the overlap between DIP-seq peaks and SMRT-seq methylation by using the bedgraph→bigwig conversion format, with fractional value attached to each methylation position. We used deepTools for plotting DIP-seq peak profiles, centered around the peak summit, for each methylation mark. Indeed, we can clearly see the co-localization of both techniques for 4mC and 6mA (see the added Supplementary Fig. 6a), where regions harboring a called peak from DIP-seq have an increase of methylated sites/fractional values called after SMRT-seq analysis. As requested, we also plotted the corresponding profile using only CpG sites within SMRT-seq 4mC methyl sites (Supplementary Fig. 6a, middle), which similarly shows an increase towards 4mC peaks.

4) The 4mC and 6mA fractional bigwig files should also be used to present metaProfiles as those shown in Figure 2a. The fractional values should also show an average increase of 4mC marks on TEs, and maybe confirm this very subtle enrichment in the TSS.

The 4mC and 6mA fractional bigwig files generated from SMRT-seq data (from point #4) were taken to represent their values across annotated TEs to confirm this enrichment. As was previously done in Figure 2c, we plotted TE metaprofiles with deepTools using different upstream/downstream regions and bin sizes. Once again, the TE insertion point (5' end) shows an overlying increase of 4mC marks, which have now been factored by fraction values (see the added Supplementary Fig. 6b).

5) The authors say that there are not many TEs in Adineta. The heatmaps seem quite data rich. So how many are shown in each plot? That's important to interpret those maps, and to draw potential differences between both strains.

We agree that the heatmaps should display information on the data density, so that the highly dense datasets (genes) could be compared with less abundant annotations (in this case, TEs). While we chose to keep the same heatmap height, we have now added the total number of annotations (genes, TEs) on the side of each heatmap in Fig. 4a-d, to provide a context for comparison and interpretation.

Minor remarks. Abstract – DNA modification is used -> "DNA modifications are used" feels more natural. Then, the second sentence with an "it" is confusing, what is the "it" referring to?

Both sentences were corrected as requested; “it” was replaced with “modifications”.

This sentence: “N4CMT adds 4mC to DNA, and its chromodomain shapes the “histone-read-DNA-write” architecture, together with a “DNA-read-histone-write” SETDB1 H3K9me3 histone methyltransferase variant preferentially binding 4mC-DNA, to maintain 4mC and silent chromatin at active transposons and tandem repeats.” Is very long and hard to follow. Furthermore, the connection of SETDB1 with 4mC is very premature to be highlighted like this in the abstract. This would be, at most, a suggestion with the current evidence.

This long sentence was split in two, with the second sentence reflecting the suggestive nature of SETDB1 findings (lines 24-28).

“Our results bring the third base modification into the eukaryotic repertoire” this is simply wrong, there are many other base modifications found in eukaryotes. E.g. J base in Trypanosomatids, hydroxymethylcytosine (and other derivatives), 5-Hydroxymethyluracil in dinoflagellates, or this one (<https://pubmed.ncbi.nlm.nih.gov/31043749/>).

Indeed, our intention was to focus on bacterial modifications rather than the numerous existing eukaryotic derivatives. We have replaced “the third base modification” with “the third bacterial modification”.

“epigenetic systems to suppress transposon proliferation”. Well, one thing is that it is marking TEs, another one is that is suppressing their proliferation, which this work does not demonstrate.

Replaced with “epigenetic systems to silence transposons”. Our work emphasizes strong correlations with transposon proliferation based on comparative analysis of multiple rotifer species, especially the TE-rich bdelloid *D. carnosus* without N4CMT which lost the 4mC-preferring SETDB1 variant, as shown in Figure 6. However, observing the actual reduction in genomic copy numbers over the evolutionary time scales would be a long-term process, which is shaped by several opposing forces and extends far beyond what is possible to achieve in the lab during an experimental study in metazoan species.

If 4mC is marking TEs, then it does not have anything to do with “regulatory networks”. Silencing a TE is not a “regulatory” network, but a silencing mechanism, a definition of “regulatory network” according to the Nature publishing group can be found here: <https://www.nature.com/subjects/regulatory-networks#:~:text=Definition,genes in a given genome.>

Although the involvement of transcription factors in regulation of expression, as required by the above definition, is strongly suggested by the concentration of epigenetic marks near the TSS (as mentioned in the discussion), it does not constitute the principal focus of the current study. We have therefore replaced “gene regulatory networks” with “gene silencing systems”, although this concept is broader and includes not only TGS, but also PTGS and nuclear organization (<https://www.nature.com/subjects/gene-silencing>).

The authors should define in the introduction what do they mean by “epigenetic”. There’s a lot of controversy in the field, therefore a clear definition of what do they mean is mandatory (since they find this a unique feature of this system).

This is a very useful suggestion, given the possible contradictory definitions. For consistency with the previous item, for the purpose of this study we keep following the definitions on the same website (<https://www.nature.com/subjects/epigenetics>), and explain in the Introduction (lines 57-59): “We focus our attention on epigenetic silencing phenomena that involve DNA and histone modifications, without expanding into broader areas involving nuclear organization or post-transcriptional silencing.”

Reviewer #4 (Remarks to the Author):

Authors have improved the paper. I think the paper can be accepted.

We are pleased that the reviewer finds our responses satisfactory.

REVIEWERS' COMMENTS

Reviewer #3 (Remarks to the Author):

The authors have addressed most of my points, here are my minor remaining points:

The authors mention that they have used "input DNA" to correct DIP-seq tracks. That's fine, but is the new Figure 2a showing the background corrected version? "IP occupancy" is a bit cryptic and I cannot guess from the Figure legend. In Figure 4e/f it says "log2 ratio" which is very evident, so it is a bit strange that two plots showing the same data type have different y-axis labels/measures.

Supplementary Figure 8 is really useful, I thank the authors for this. The first example is really clear, with a very nice cluster of 4mC and 6mA sites. The two other examples are a bit more sparse on 4mC, were they chosen for a particular reason? Since there are not expectations on how 4mC might look like in any eukaryotic genome, I guess this is fine, but for the readers it would be nice to know the rationale (first example being the clearest in the genome and the other two being the 2nd and 3rd? or chosen for other reasons?).

The bigwig tracks made for 4mC and 6mA would be important to upload to a public repository (e.g. the GEO? Or Figshare/github/etc), since the analysis of PacBio is not straightforward and these data might be useful for other researchers interested in the findings described in this manuscript.

RESPONSE TO REVIEWERS' COMMENTS

Reviewer #3 (Remarks to the Author):

The authors have addressed most of my points, here are my minor remaining points:

The authors mention that they have used “input DNA” to correct DIP-seq tracks. That’s fine, but is the new Figure 2a showing the background corrected version? “IP occupancy” is a bit cryptic and I cannot guess from the Figure legend. In Figure 4e/f it says “log₂ ratio” which is very evident, so it is a bit strange that two plots showing the same data type have different y-axis labels/measures.

For ChIP-seq data, we used input DNA as chromatin control in both strains for background correction using MACS2. Using bamCompare with both the treatment and the input control, we obtained the log₂ ratio between the two (Figure 4e/f). In 4mC and 6mA DIP-seq experiments, we used MACS v. 1.4.2, rather than MACS2, for peak calling in DIP-seq data, since it is considered effective in capturing the local genomic sequence biases from a ChIP-seq sample alone, in lieu of the control sample (Zhang et al. 2008). Thus, Figure 2a is showing IP occupancy, as there is no input to derive a log₂ ratio from DIP-seq data.

Supplementary Figure 8 is really useful, I thank the authors for this. The first example is really clear, with a very nice cluster of 4mC and 6mA sites. The two other examples are a bit more sparse on 4mC, were they chosen for a particular reason? Since there are not expectations on how 4mC might look like in any eukaryotic genome, I guess this is fine, but for the readers it would be nice to know the rationale (first example being the clearest in the genome and the other two being the 2nd and 3rd? or chosen for other reasons?).

Contig As785 is by far one of the highest in methylation density which is observed; it covers a tandem repeat (shown in Supplementary Fig. 7a) located between *Athena* elements. Other contigs with 4mC and 6mA clusters normally show lower density of methylated sites, as illustrated in two other examples showing DNA TEs, which were not presented in earlier circos plots. Contig 1073 represents an under-annotated transposon region (Supplementary Table 11), which contains a potentially active, polymorphic insertion of *Chapaev* DNA TE present only in PacBio, but not in Illumina data. Contig1534 is another example of association between 4mC methylation marks and TE insertion (while the 4mC marks are indeed more sparse, the 6mA marks are entirely lacking). This information has been included into the figure legend.

The bigwig tracks made for 4mC and 6mA would be important to upload to a public repository (e.g. the GEO? Or Figshare/github/etc), since the analysis of PacBio is not straightforward and these data might be useful for other researchers interested in the findings described in this manuscript.

BigWig (BW) tracks for 4mC and 6mA have been added to the GEO dataset GSE140050 [<https://www.ncbi.nlm.nih.gov/geo/query/acc.cgi?acc=GSE140050>], which has been made publicly available.